# Engineering *Bacillus subtilis* for the formation of a durable living biocomposite material

Sun-Young Kang [1,2,4], Anaya Pokhrel [1,2,4], Sara Bratsch [1,2,4], Joey J. Benson[3], Seung-Oh Seo[1,2], Maureen B. Quin[1,2], Alptekin Aksan [2,3] & Claudia Schmidt-Dannert [1,2 ✉]

Engineered living materials (ELMs) are a fast-growing area of research that combine approaches in synthetic biology and material science. Here, we engineer *B. subtilis* to become a living component of a silica material composed of self-assembling protein scaffolds for functionalization and cross-linking of cells. *B. subtilis* is engineered to display SpyTags on polar flagella for cell attachment to SpyCatcher modified secreted scaffolds. We engineer endospore limited *B. subtilis* cells to become a structural component of the material with spores for long-term storage of genetic programming. Silica biomineralization peptides are screened and scaffolds designed for silica polymerization to fabricate biocomposite materials with enhanced mechanical properties. We show that the resulting ELM can be regenerated from a piece of cell containing silica material and that new functions can be incorporated by co-cultivation of engineered *B. subtilis* strains. We believe that this work will serve as a framework for the future design of resilient ELMs.

[1] Department of Biochemistry, Molecular Biology & Biochemistry, University of Minnesota, Minneapolis, MN 55455, USA. [2] BioTechnology Institute, University of Minnesota, St. Paul, MN 55108, USA. [3] Department of Mechanical Engineering, University of Minnesota, Minneapolis, MN 55455, USA. [4]These authors contributed equally: Sun-Young Kang, Anaya Pokhrel, Sara Bratsch. ✉email: schmi232@umn.edu

The design of cells capable of producing self-organizing, biocomposite materials has the potential to enable the fabrication of new types of functional living materials with the ability of self-fabrication and self-repair. Engineered living materials (ELMs) are therefore a new and fast-growing area of research that combines approaches in synthetic biology and material sciences[1–4]. The fabrication of most ELMs, however, has so far largely relied on physical methods for incorporating a living component in an external material[4–7]. Engineering of 'truly' living materials where the living component actively facilitates material fabrication and organization is much more challenging. True ELMs have been created by engineering *Escherichia coli* to produce an extracellular matrix from curli fibers[8–25]. Other types of extracellular matrices for ELM fabrication were created from secreted bacterial cellulose to embed microbial cells[26,27] or from elastin-like polypeptides to attach *Caulobacter* cells via their protein S-layers[28]. Among these examples, the secretion of a polypeptide-based scaffolding system or matrix offers greater control over material assembly and functionalization due to the genetic programmability of polypeptide structures and functions.

Despite the advances in ELM design, the diversity of different ELM types capable of autonomous self-fabrication and regeneration is still small. In addition, ELM fabrication is currently limited to common chassis organisms that lack the resilience and long-term viability capabilities to withstand conditions outside of the laboratory. Here, we sought to broaden the ELM landscape by engineering a resilient ELM biocomposite that uses the spore-forming bacteria *Bacillus subtilis* as its living component for the secretion of self-assembling protein scaffolds for cell cross-linking and silica biomineralization.

*B. subtilis* is an industrially used GRAS (Generally Recognized as Safe) bacteria known to have excellent protein secretion capabilities. Importantly, it forms spores that remain viable for a long time and allow the bacteria to survive extreme conditions[29–31]. *B. subtilis* cells will therefore be able to enter a dormant spore state in our ELM under unfavorable environmental conditions, allowing it to persist until favorable conditions induce germination and cell revival. This was recently demonstrated by bioprinting of *Bacillus* spores in an agarose matrix[6]. In addition, because spores contain the genetic information that was programmed into engineered vegetative cells, living materials may be autonomously fabricated at the sites of use from stored spores. For the secretion of a self-assembling protein matrix, we chose a highly robust 2D-scaffolding system from bacterial microcompartment shell proteins that we have characterized and engineered for the attachment of different cargo proteins[32–35]. We rationalized that 2D-scaffold forming proteins rather than the commonly engineered bacterial biofilm-associated amyloid fibers (e.g. CsgA from *E. coli* or TasA from *B. subtilis*[10]) will form different matrix architectures and surfaces for functionalization. Demonstration of extracellular matrix formation from nonamyloid protein building blocks will also lay the foundation for the design of new types of ELM matrices from the many other protein building blocks currently assembled into functional bionanomaterials[36]. Our 2D-scaffolds self-assemble from hexameric units of the bacterial microcompartment shell protein EutM. EutM scaffold building blocks are highly amenable to engineering and tolerate N- and C-terminal fusions, including a C-terminal SpyCatcher domain for the covalent linkage of SpyTag-modified proteins to scaffolds[34,35]. We therefore rationalized that *B. subtilis* could be engineered to secrete EutM-SpyCatcher (EutMSpyC) building blocks and attach itself covalently via iso-peptide bond formation by the SpyTag-SpyCatcher system to the formed scaffolds to become both a structural and the living component of the formed material. Further, by engineering the scaffolds to display a biomineralization peptide (EutM-BM),

scaffold mineralization could then yield a durable biocomposite material for future development into functional coatings or plasters. As a proof-of-concept, we chose silica, as one of the most abundant earth minerals that is inexpensive and the main ingredient of many building materials. A number of proteins are also known to control the biomineralization orthosilicic acid $Si(OH)_4$ (the soluble form of silica), which can be used as a source for biomineralization peptides[37].

In this work, we first optimize secretion by *B. subtilis* of our EutM-SpyCatcher scaffold building blocks, which are not normally secreted and in addition, rapidly self-assemble both in vivo and in vitro into large structures[32–35,38–40]. We then show that *B. subtilis* can be engineered to display SpyTags on clusters of polar flagella for cell attachment and cross-linking of EutM scaffolds that constitute the protein matrix of our ELM. To allow our production strain to remain a structural component of the formed material even in its dormant state, we create an endospore-forming strain. Screening of known silica biomineralization peptides then allows us to design scaffolds for silica polymerization to fabricate biocomposite materials with enhanced mechanical properties. Finally, we confirm that our ELM can be regenerated from a piece of cell containing silica material and that new functions can be incorporated into the material simply by co-cultivation of engineered *B. subtilis* strains. Figure 1 illustrates this workflow from strain engineering to material fabrication and regeneration. We believe that this work will serve as a framework for the future design of resilient ELMs as functional, self-healing materials for use as coatings and plasters that can respond to external stimuli due to the functions provided by the engineered cells in such materials.

## Results and discussion

**Identification of growth conditions for ELM fabrication.** Several factors need to be considered for the fabrication of a living silica biocomposite material. *Bacillus* growth conditions must be identified that are permissible for silica biomineralization and preferably, do not create an overly discolored material for aesthetic and functional (for example for sentinel or stealth applications) reasons. These conditions must also support scaffold building block expression and secretion during exponential to early stationary phase growth when cells are flagellated and have not yet initiated sporulation. Growth conditions for *B. subtilis* 168 (referred to as WT) were therefore evaluated using two different media; a Luria Bertani (LB) and a clear, buffered Spizizen minimal medium (SMM). *Bacillus* cells were transformed with a plasmid (pCT-empty) derived from our previously developed cumate (*p*-isopropyl benzoate) inducible *Bacillus* expression vector (Supplementary Data 1)[41]. Bacterial growth, media pH, flagella presence, and sporulation were then analyzed for 4 days in 24 h intervals at 20 and 30 °C (Supplementary Figs. 1, 2, and 3). Although pH is not typically considered in culture conditions, silica polymerization is optimal between pH 6.0 and 8.0 and at pH levels above and below this range polymerization is inhibited[37]. During growth of *B. subtilis*, the culture medium turns increasingly alkaline by switching from sugar to amino acid catabolism - releasing ammonia from tryptophan (SMM) or hydrolyzed amino acids from tryptone and yeast extract (LB) (along with the secretion of alkaline proteases)[42]. In buffered SMM at 20 °C, the pH was maintained around 7.0 until 48 h of growth, while under all other conditions tested and during longer growth, media pH quickly increased above 7.0 (Supplementary Fig. 1). Peritrichous flagella (uniformly distributed flagella) were observed through 72 h of growth without significant sporulation (Supplementary Fig. 2). Buffered SMM and growth for 48 h at

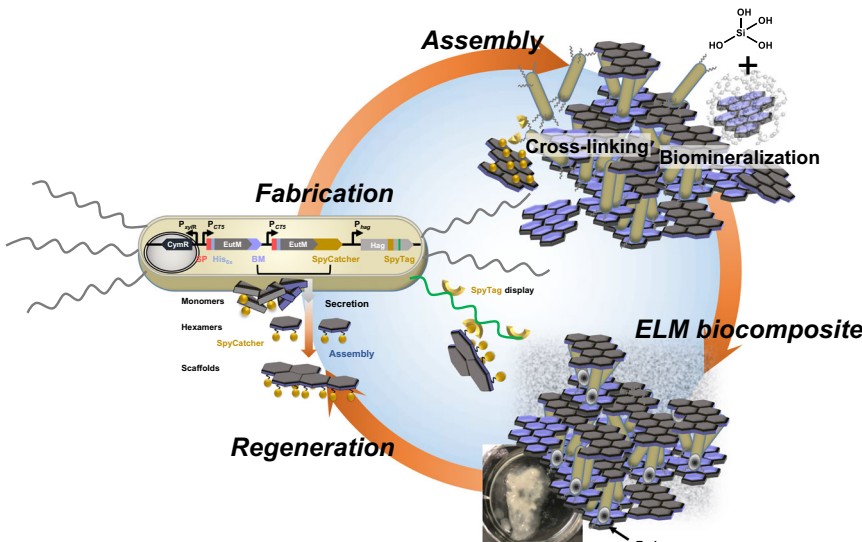

**Fig. 1 Formation of a living silica biocomposite material.** *B. subtilis* is designed to secrete a protein matrix from self-assembling EutM scaffold building blocks. Cells are engineered to display a SpyTag on clustered, polar flagella for covalent cross-linking via isopeptide bond formation with SpyCatcher domains fused to secreted scaffold building blocks. Scaffold building blocks are genetically fused with peptides to enhance silica polymerization on scaffolds. Finally, cells are engineered to retain spores as endospores. This allows cells to persist as structural components of the material and creates a durable biocomposite material that can be regenerated from a piece of silica material containing cells. (BM biomineralization peptide, SP, secretion signal peptide).

20 °C therefore appeared to be optimal for our ELM production by engineered *B. subtilis*.

We then repeated these growth experiments with *B. subtilis* WT transformed with a plasmid (pCT-EutMSpyC) for the secretion of the scaffolding protein EutM-SpyCatcher modified with the N-terminal secretion peptide SacB from *B. subtilis* (Fig. 1 and Fig. 2a). As before, pH increased above 7.5 regardless of medium at 30 °C after 24 h in LB and after 48 h in SMM. (Supplementary Fig. 1). Again, only cultures grown in SMM at 20 °C maintained a stable pH around 7.0 for 48 h and even until 96 h after induction of protein expression. Notably, growth was significantly reduced at 20 °C in SMM and LB during 48 h after induction compared to the empty vector control cultures. This was most notable in SMM at 20 °C (Fig. 2b), suggesting that EutM-SpyCatcher protein overexpression and secretion may burden cells, slowing cell growth. Like the empty vector cultures above, EutM-SpyCatcher expressing cultures had peritrichous flagella through 72 h of growth, with the most flagella present until 48 h (Supplementary Fig. 3a). However, spores were already observed after 48 h of growth at 20 °C in LB media, indicating cells were stressed under this condition (Supplementary Fig. 3b). Based on these results, we chose to perform all subsequent cultivations for ELM production in SMM at 20 °C for 48 h as our standard condition.

**Establishing EutM scaffold building block secretion.** The majority of industrially produced recombinant proteins by *B. subtilis* are extracellular enzymes (e.g. hydrolases) and achieving high-level secretion typically requires the identification of efficient secretion signal peptides to replace their native secretion signals[43–45]. In contrast, our scaffolding protein EutM is cytoplasmic that also rapidly self-assembles into scaffolds when overexpressed in *E. coli*[32–35,38–40]. We therefore screened different signal peptides and investigated secretion of EutM-SpyCatcher by *Bacillus*. In addition to SacB, we tested four other, commonly used secretion peptides (XynA, YngK, CelA, and LipA)[46–50] by fusing them to the N-terminus of our scaffold protein (Supplementary Fig. 4a).

Each recombinant protein was expressed in *B. subtilis* WT in SMM, at 20 °C for 48 h as the optimal condition confirmed above. Protein secretion levels were then compared between the different cultures by SDS-PAGE analysis. We analyzed total protein in the culture, and after centrifugation of cultures, protein in the resulting culture supernatant and pellet. However, a very low amount of secreted protein was observed in the supernatant compared to visible bands in the pellet, and we suspected that secreted and scaffolded EutM proteins were pelleting together with the cells. We therefore solubilized scaffold proteins from pelleted cells with 4 M urea, which we have used previously to solubilize EutM scaffolds produced by *E. coli*[33,35]. At this concentration, urea does not lyse cells which we verified by comparing total protein released from completely lysed cells after urea treatment and protein solubilized from sediment (see Supplementary Fig. 5). We compared by SDS-PAGE three fractions as shown in Fig. 2c: (A) culture supernatants (concentrated by trichloroacetic acid (TCA)), (B) urea supernatant after scaffold solubilization from pellets, and (C) total protein released from remaining pellet after urea solubilization and cell lysis. Only the SacB secretion signal peptide yielded significant yields of secreted protein in the culture supernatant and the solubilized fraction (Fig. 2d). Only small amounts of secreted EutM protein (only after 25x concentration) could be detected in culture supernatants with the other four signal peptides (Supplementary Fig. 4b). Compared to cultures expressing SacB-EutM-SpyCatcher, no scaffolds sedimented in the other cultures secreting low levels of EutM protein (Supplementary Fig. 4b). From these results, we concluded that SacB is the best secretion signal peptide for EutM-SpyCatcher.

**Engineering *B. subtilis* cells as integral material component.** With scaffold building block secretion confirmed, we next set out to engineer our host strain to persist as a component within the matrix it fabricates by providing structure to the material and for material regeneration when favorable growth conditions are provided (Fig. 1). *B. subtilis* spores are extremely durable and can remain dormant indefinitely until proper environmental conditions return whereupon the spores will germinate[29,31]. During

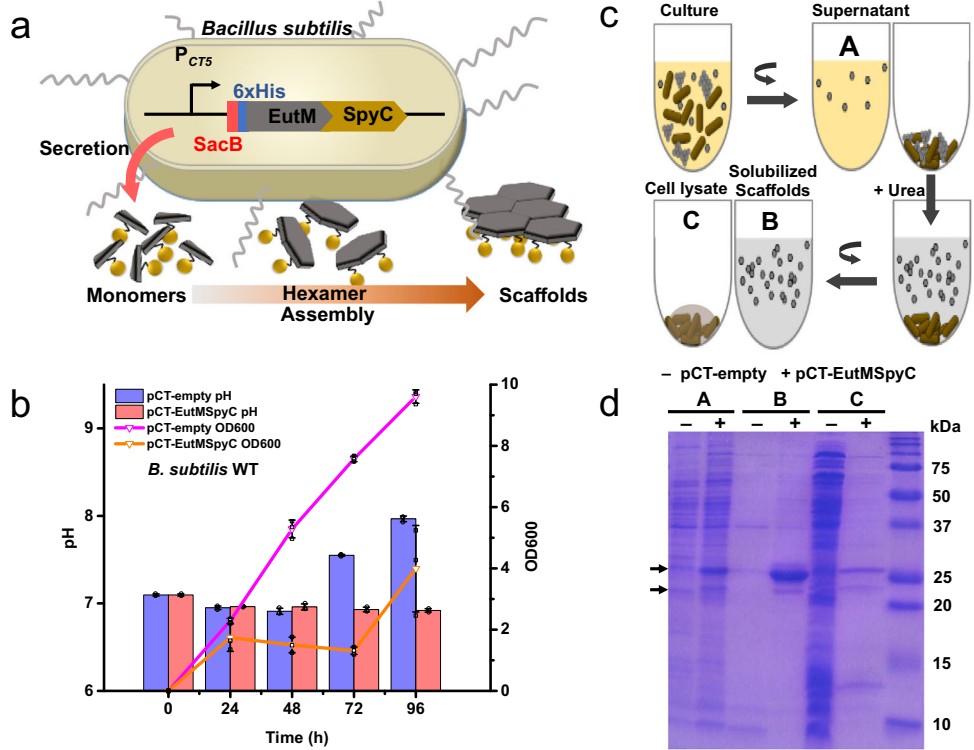

**Fig. 2 Establishment of self-assembling scaffold building block secretion. a** Self-assembling EutM scaffold building blocks containing a C-terminal SpyCatcher (SpyC) domain and a SacB secretion signal sequence followed by a His-tag are expressed and secreted by *B. subtilis* under the control of a cumate inducible promoter. **b** Growth and pH of cultures expressing EutMSpyC was followed for 96 h in SMM at 20 °C and compared to empty plasmid control cultures. Data are shown as mean values ± SD and error bars represent the standard deviations of three independent biological replicate cultures. Colored bars and lines represent mean values. Black symbols represent data points and error bar. **c** Self-assembling scaffolds settle with cells, requiring their solubilization from cell pellets by a gentle wash with 4 M urea (see Methods for details). **d** Protein expression and secretion after 48 h of cultivation were analyzed by SDS-PAGE where: (A) is the culture supernatant (10-fold concentrated by TCA precipitation), (B) urea supernatant after scaffold solubilization, and (C) resulting cell pellet after lysozyme treatment for analysis of remaining protein. Arrows indicate EutM-SpyCatcher bands. Expected sizes are 24.4 kDa and 21.1 kDa with and without SacB secretion signal peptide, respectively. The shown data is representative of three independent biological replicate cultures. Source data are provided as a Source Data file.

sporulation, endospores are formed in the mother cell which then undergoes lysis for spore release. To retain *Bacillus* cells as structural material components while also allowing spore formation for long-term viability of our material, we deleted the cell wall autolysin LytC (N-acetylmuramoyl-L-alanine amidase) in *B. subtilis*[51] by in-frame deletion of *lyt*C to generate a *B. subtilis* Δ*lyt*C strain (Fig. 3a and Supplementary Fig. 6).

As our objective is to link *B. subtilis* cells to secreted EutM-SpyCatcher scaffold via SpyTag display on its flagella (Fig. 1), we also introduced a second in-frame deletion of *flh*G (Fig. 3a and Supplementary Fig. 6) to change the flagellation pattern on cells from peritrichous to polar flagella clusters. Normally, *B. subtilis* produces flagella around the midcell and lacks flagella at the poles[52]. The protein FlhG (also called YlxH, MinD2, FleN, and MotR) is essential to this flagellation pattern[53,54]. The deletion of *flh*G results in the aggregation of basal bodies at the polar ends of cells[52]. We reasoned that reducing flagellation density and directing flagella to the poles will enable more spaced-out cross-linking on scaffolds and less 'clumping' of cells (which we later serendipitously discovered appears to be deleterious for *B. subtilis*, see below).

The resulting double deletion strain *B. subtilis* Δ*lyt*C Δ*flh*G was then transformed with expression vectors (pCT-empty, pCT-EutMSpyC) for confirmation of expected cell phenotypes and scaffold building block secretion (Fig. 3b–d). Comparison of flagella pattern between wild-type *B. subtilis* and the double deletion strain confirmed the presence of clustered polar flagella

and cell wall enclosed endospores in the deletion strain (Fig. 3b). Clustered polar flagella were most prevalent at 48 h or less after induction, and the most flagella both for the wild-type and the double deletion strain were observed in cultures grown at 20 °C (Supplementary Figs. 2a, 3a, 6d). Unlike the wild-type strain, no free spores were released by the autolysin deficient Δ*lyt*C strain in SMM (Supplementary Figs. 2b, 3b, 6c). Endospore formation by *B. subtilis* Δ*lyt*C Δ*flh*G harboring pCT-empty and pCT-EutMSpyC was not observed until 72 h of growth under the optimal growth conditions (SMM, 20 °C) identified above (Fig. 3b).

Following the confirmation of clustered flagella and no sporulation during growth in SMM at 20 °C for 48 h after induction, we next evaluated growth and protein expression under these conditions (Fig. 3c, d). Interestingly, the growth rate of the double deletion strain was steadier and higher than the wild-type strain when transformed with either the empty control plasmid or pCT-EutMSpyC (compare Fig. 2b and Fig. 3c). The pH of the culture was also maintained at around 7.0 for 48 h. Expression and secretion of scaffold building blocks by the engineered strain was unchanged from the wild-type strain (Fig. 3d, compare to Fig. 2d) despite reaching a slightly higher optical density after 48 h.

We noticed that the secreted EutM-SpyCatcher protein appears as a major higher (24.4 kDa) and a minor lower (21.1 kDa) molecular weight band on SDS-PAGE gels in all three analyzed fractions (culture supernatant, urea solubilized scaffolds and lysed

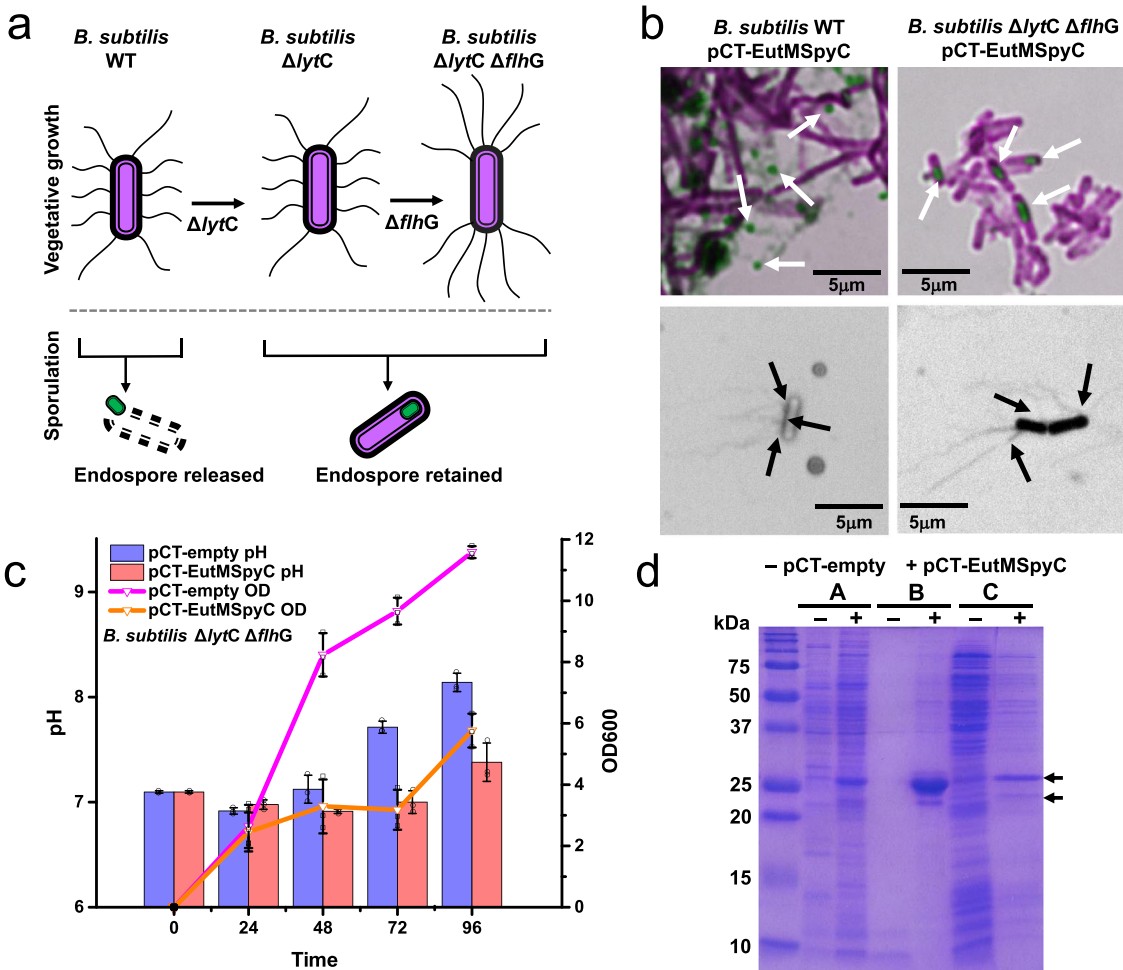

**Fig. 3 Development of a *Bacillus* strain for durable material fabrication. a** Construction of a *B. subtilis* Δ*lyt*C Δ*flh*G strain deficient in spore release and exhibiting polar, clustered flagella was achieved by deleting the autolysin LytC and the flagella basal body protein FlhG. **b** Sporulation and flagella phenotypes of *B. subtilis* strains transformed with pCT-EutMSpyC were compared to confirm endospore and polar flagella phenotype under expression conditions. Representative samples show spore release for the wild-type strain (LB, 48 h, 20 °C shown) and endospore formation by the engineered strain (SMM, 72 h, 20 °C shown) (see also Supplementary Figs. 2, 3 and 6). Peritrichous flagella are observed for the wild-type strain (LB, 48 h, 20 °C shown) under all tested growth conditions through 72 h and flagellation decreases after 48 h (see also Supplementary Figs. 2 and 3). Clustered polar flagella are observed in the engineered strain through 48 h of growth (SMM, 48 h, 20 °C shown) (see also Supplementary Fig. 6). Samples were stained with either malachite green and safranin red (top) or RYU (bottom) and examined by light microscopy at 100x. Spore images had red replaced with magenta and flagella images were converted to grayscale. Images shown are representative three independent biological replicate cultures. **c** B. subtilis Δ*lyt*C Δ*flh*G transformed with the pCT-empty (control) or pCT-EutMSpyC plasmid (SMM, 20 °C) show growth characteristics comparable to the *B. subtilis* wild-type strain in Fig. 2. Data are shown as mean values ± SD and error bars represent the standard deviations of three independent biological replicate cultures. Colored bars and lines represent mean values. Black symbols represent data points and error bar. **d** Expression and secretion of EutM-SpyCatcher by the engineered *B. subtilis* Δ*lyt*C Δ*flh*G strain was verified in SMM medium at 20 °C after 48 h of induction by SDS-PAGE. Arrows indicate EutM-SpyCatcher bands that are not present in the empty plasmid control. A strong band corresponding to urea solubilized EutM-SpyCatcher scaffolds (B) from co-pelleted cells confirm comparable protein expression and secretion levels of the engineered and the wild-type strains (Fig. 2). Samples (A) and (C) corresponding to 10-fold concentrated culture supernatant and cell lysate after scaffold solubilization also show comparable protein bands to the wild-type strain. Shown data is representative of three independent biological replicate cultures. Source data are provided as a Source Data file.

cell pellet, Figs. 2d and 3d). We previously observed that EutM proteins do not migrate according to size and/or as double bands[33,38,40]. Here, we wondered whether the differences in apparent molecular weights might be due to uncleaved SacB signal peptide. Secretion of native proteins with uncleaved Sec signal sequences into culture supernatants has been documented for bacteria[55]. LC-MS analysis showed that indeed the signal peptide was not cleaved from either EutM protein band (Supplementary Fig. 7). This suggests that EutM likely folds rapidly upon emerging from the Sec translocase complex[43,44], preventing access for the signal peptidase and pulling the folding protein into the extracellular space. EutM and its homologs are

known to rapidly fold and self-assemble in vitro and in vivo to form higher-order structures[32–35,38–40]. In contrast, the release and folding of the *B. subtilis* levansucrase, which is the source of the SacB secretion signal sequence, has been shown to be a slow process[56].

To determine whether the remaining SacB signal peptide effects EutM-SpyCatcher assembly and to compare EutM-SpyCatcher protein assembly in SMM medium to previous observations in Tris buffer, we purified uncleaved and secreted protein from recombinant *B. subtilis* Δ*lyt*C Δ*flh*G cultures and His-EutM-SpyCatcher without signal sequence from recombinant *E. coli* cultures. After metal affinity chromatography and dialysis

into SMM, scaffolds were observed by transmission electron microscopy (TEM). In comparison to the negative control of SMM which shows many nonhomogenously sized and poorly stained structures (Supplementary Fig. 8a, e), scaffolds isolated from *E. coli* formed rolled-up tubes (Supplementary Fig. 8b) which have been observed previously for EutM proteins expressed in *E. coli*[33]. *Bacillus* secreted protein assembled into fewer large, rolled-up tubes and smaller, spherical structures (that stain as donut-like structures) as well as sheets and less-ordered coral-like scaffolds (Supplementary Fig. 8c, d, f–h). These structures have also been previously observed for EutM homologs[33,40]. This greater variety of scaffold assemblies may be due to the SacB sequence, modifying surface charges and assembly. These results confirm that *B. subtilis* secreted EutM-SpyCatcher forms scaffolds and that the SMM growth medium does not inhibit self-assembly.

**Linking *B. subtilis* cells to secreted EutM-SpyCatcher scaffolds by SpyTag display**. We next set out to engineer the display of a SpyTag on flagella for cell attachment to secreted scaffolds (Fig. 1). For characterization of flagella SpyTag display, a cysteine variant (T209C) of the flagellin Hag protein was first created for labeling of flagella filaments with a sulfhydryl-specific (maleimide) dye[57]. The modified flagellin gene was then expressed from its native promoter on a plasmid in wild-type *B. subtilis* and the ΔlytC ΔflhG double deletion strain. The assembled flagella filaments are therefore composed of both engineered and native flagellin monomers. The recombinant Hag[T209C] protein was then modified to include a SpyTag domain at nine different positions chosen inside the variable D2/D3-region of the flagellin protein based on the assumption that mutations in these locations would least likely interfere with filament assembly and function (Fig. 4a, b)[58]. Four Hag insertions (bp positions 399, 426, 459, 537) were deleterious for both *B. subtilis* strains, meaning that either no transformants could be obtained or transformants contained missense mutations or deletions in the modified *hag* gene of the recovered plasmids (Fig. 4a). Fluorescence microscopy (Fig. 4c) of the remaining mutants revealed flagella located at the mid cell for *B. subtilis* WT and poles for *B. subtilis* ΔlytC ΔflhG which adopt different filament conformations (Figs. 4d, S9a). Wild-type flagella form a left-handed helix[59], which was retained for insertions at positions 612 and 645 in Hag[T209C]. Curly, right-handed forms (with half the pitch and amplitude) were observed for flagella expressing SpyTags at positions 555 and 588. SpyTag insertion at position 588 also resulted in shortened flagella compared to unmodified flagella (Fig. 4d). One strain expressing Hag[T209C] with a SpyTag at position 492 had both normal and curly flagella (Supplementary Fig. 9a).

Functional display of the SpyTag on *B. subtilis* flagella by the different insertion mutants was then evaluated by adding purified red fluorescent tdTomato-SpyCatcher protein (Fig. 4c) to recombinant *B. subtilis* WT and ΔlytC ΔflhG cultures that were then grown overnight for flagella expression. No attachment of tdTomato-SpyCatcher to flagella of strains with Hag[T209C] SpyTag insertions at positions 492, 555, and 612 was observed by fluorescence microscopy (Fig. 4e, Supplementary Fig. 9b). Some red fluorescent labeling was observed for SpyTag display at position 645 (Supplementary Fig. 9b), but the best attachment of tdTomato-SpyCatcher was achieved on flagella that display a SpyTag at position 588 (Hag[T209C]::SpyTag[588]), despite the shorter than wild-type flagella formed by the recombinant strains (Fig. 4e).

We then confirmed that Hag[T209C]::SpyTag[588] modified flagella on *B. subtilis* ΔlytC ΔflhG are capable of forming isopeptide bonds for covalent attachment to EutM-SpyCatcher proteins

secreted during cultivation. For this, two *B. subtilis* ΔlytC ΔflhG strains - one expressing Hag[T209C]-SpyTag[588] and the other EutM-SpyCatcher – were co-cultured under the identified conditions for scaffold building block secretion but extended to 96 h of induction (SMM, 20 °C, 96 h induction). Flagella were then isolated from cells and analyzed by SDS-PAGE, confirming the presence of a higher-molecular weight protein complex (55 kDa) that was not present in the control (Fig. 4f). This band is weaker compared to the major, genome encoded flagellin band as the modified flagella only contain a small number of engineered, SpyTag-displaying flagellin monomers. Based on these results, we therefore chose Hag[T209C]::SpyTag[588] expression for cross-linking *B. subtilis* ΔlytC ΔflhG cells to EutM-SpyCatcher scaffolds.

**Engineering of silica biomineralization**. Our next objective was to modify EutM scaffolds such that they would biomineralize silica for the fabrication of a resilient living biocomposite material (Fig. 1). We selected four known silica biomineralization peptides (R5, CotB1p, SB7, and Synthetic[60–64] for C-terminal fusion to His-EutM (Supplementary Fig. 10a). We confirmed scaffold formation with the fusion proteins overexpressed and purified from *E. coli* and characterized silica precipitation. We intentionally left the His-Tag on all of our scaffold building blocks as histidine residues are enriched in proteins involved in natural silica biomineralization processes such as e.g. in the histidine-rich glassin protein present in the siliceous skeletons of sponges[65].

Silica precipitation by the biomineralization peptide modified His-EutM and the unmodified His-EutM scaffolds was then quantified using a spectroscopic molybdate blue assay[66] (Supplementary Fig. 10a). The CotB1p (hereafter referred to as CotB) peptide fusion was most efficient at precipitating silica at $7.44 \pm 0.36$ mmol/g protein and was also the most cationic peptide, rich in arginine residues. As expected, His-EutM scaffolds also precipitated silica at $1.98 \pm 0.01$ mmol/g protein. The structures (scaffolds) formed by the fusion proteins and His-EutM were then visualized by SEM (Fig. 5a and Supplementary Fig. 10b, left column). His-EutM formed tubular nanostructures that appeared to be rolled-up tubes fused together at multiple locations. In contrast, His-EutM-R5 formed globular structures fused into chains, while His-EutM-CotB formed similar scaffolds but with a greater surface roughness. Fusion of the SB7 and Synthetic peptide to EutM resulted in the formation of dense scaffolds comprised of fused together globular and poorly-formed tubular structures (Supplementary Fig. 10b).

Silica precipitation on scaffolds was investigated by incubating 1 mg/mL scaffolds for 2 h at room temperature with 100 mM silica. SEM imaging of scaffolds (Fig. 5a, center and right) showed that silica addition resulted in an increase in scaffold surface roughness, surface porosity, and the presence of silica nanoparticles fused to the scaffolds which are characteristics of silicification (Fig. 5a and Supplementary Fig. 10b, center and right columns). Silica nanoparticle formation was most prominent on the surface of His-EutM scaffolds, suggesting that biosilicification may have been nucleated in the solution phase by these proteins. In contrast, a significant increase in surface roughness seen for His-EuM-R5 and His-EutM-CotB scaffolds suggests surface catalyzed biomineralization by these structures. TEM imaging of silica precipitation on TEM grids coated with purified His-EutM and His-EutM-CotB proteins also showed the deposition of a more granular layer on His-EutM-CotB coated surfaces compared to His-EutM (Supplementary Fig. 10c), despite the formation of curved, fibrillar structures by His-EutM-CotB as opposed to the uniform structures made by His-EutM.

Based on these results, we chose the CotB biomineralization peptide for fusion to scaffold building blocks for secretion by *B.*

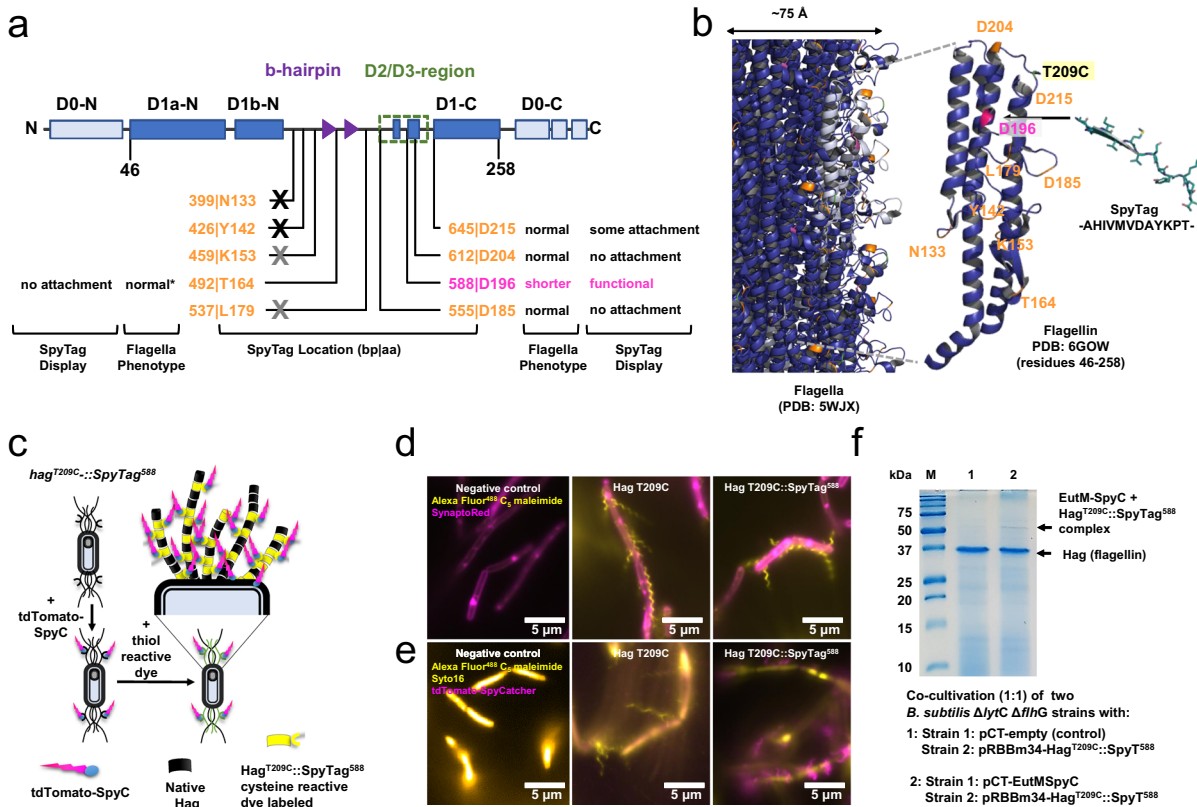

**Fig. 4 Engineering of flagella for SpyTag display. a** A SpyTag sequence was inserted at different locations in the hypervariable regions of the flagellin protein (encoded by *hag*). Shown are conserved domains, including the solvent-exposed D2/D3 domains[58]. The SpyTag insertions (orange & magenta labels indicate bp and residue locations, magenta denotes functional site) had different impacts on flagella phenotypes and SpyTag display in both *B. subtilis* 168 (WT) *and B. subtilis* Δ*lyt*C Δ*flh*G expressing plasmid born *hag*$^{T209C}$::*SpyTag*$^{xxx}$. Except for less curved flagella observed for *B. subtilis* 168 compared to normal (indicated by *) flagella for *B. subtilis* Δ*lyt*C Δ*flh*G, phenotypes were the same between the two strains. Several *hag* insertion mutants could not be transformed into either strain and seem to be lethal (black cross) when expressed in trans of the genomic *hag* copy. Gray cross: *hag* mutants could only be transformed into *B. subtilis* Δ*lyt*C Δ*flh*G but recovered plasmids were mutated. No or some attachment: modified flagella could not or only be weakly labeled with tdTomato SpyCatcher. **b** Flagellar filament and flagellin structures[82] show locations of the SpyTag insertions (orange & magenta, corresponding to labels in **a**) and T209C mutation (yellow box) for dye staining. SpyTag structure (PDB: 4MLI) and sequence are shown for reference. **c** Flagella assemble from genomic flagellin and plasmid born flagellin (Hag$^{T209C}$SpyTag$^{xxx}$) subunits. Engineered flagellins are stained with a cysteine reactive dye (Alexa Fluor$^{TM}$ 488 C$_5$ maleimide) and functional SpyTag display was tested by growing cultures with purified tdTomato-SpyCatcher protein. **d** Fluorescence imaging of *B. subtilis* Δ*lyt*C Δ*flh*G transformed with pCT-empty (control), pRBBm34-Hag$^{T209C}$ or pRBBm34-Hag$^{T209C}$::SpyT$^{588}$ show that expression of *hag*$^{T209C}$::*SpyTag*$^{588}$ results in shortened, clustered, polar flagella. Cells are stained and images false-colored as indicated. See Supplementary Fig. 9 for other SpyTag positions. **e** Functional SpyTag display by Hag$^{T209C}$::SpyTag$^{588}$ was confirmed by growing strains in the presence of purified tdTomato-SpyCatcher. (Images generated for each strain in **d** and **e** are representative of cultures selected from three biological replicate cultures.) **f** In situ attachment of secreted EutM-SpyCatcher to SpyTags displayed on flagella of *B. subtilis* Δ*lyt*C Δ*flh*G was verified by co-cultivation. SDS-PAGE analysis of extracted flagella shows a faint band with the expected size of 55 kDa for a EutM-SpyCatcher (21.1 kDa) and Hag$^{T209C}$::SpyTag$^{588}$ (34.1 kDa) protein complex in co-cultures secreting EutM-SpyCatcher. Genomic flagellin (32.6 kDa) is the major protein. Shown data is from a single experiment. Source data are provided as a Source Data file.

*subtilis*. An expression construct was made where the SpyCatcher domain in pCT-EutMSpyC was replaced with CotB to yield plasmid pCT-EutMCotB, replicating the *E. coli* expressed His-EutM-CotB with the additional SacB signal sequence (Fig. 5b). EutM-CotB expression and secretion by *B. subtilis* Δ*lyt*C Δ*flh*G transformed with pCT-EutMCotB (SMM, 20 °C, 48 h induction) was confirmed by SDS-PAGE (Fig. 5c, lane 3). As in the case of EutM-SpyCatcher, the SacB signal sequence appears to be uncleaved for EutM-CotB based on the size of the observed protein band (16.8 kDa vs 13.8 kDa for the uncleaved and cleaved proteins, respectively). Secretion of EutM-CotB however was much lower than EutM-SpyCatcher, probably due to the added positive charged by the CotB peptide fusion[67]. Yet, despite the low amount of secreted EutM-CotB, cultures of *B. subtilis* strains transformed with pCT-EutMCotB significantly increased silica gel formation at low silica concentrations compared to empty vector

control cultures (Supplementary Fig. 11). After 48 h induction of protein expression (SMM, 20 °C), samples of cultures were incubated with different concentrations of silica (50–500 mM) and silica gel formation followed for 24 h at 20 °C. With 100 mM silica added, the EutMCotB expressing cultures solidified into a gel, while the control cultures only formed a liquid soft gel (Supplementary Fig. 11).

We then proceeded with combining the expression and secretion of our EutM-SpyCatcher building block for cell cross-linking with the EutM-CotB scaffold building block for enhanced biomineralization, yielding the expression vectors as shown in Fig. 5b. For SpyTag display on flagella of *B. subtilis* Δ*lyt*C Δ*flh*G, a plasmid for co-expression of EutM scaffold building blocks with the modified flagellin subunit Hag$^{T209C}$::SpyTag$^{588}$ was also constructed. Each expression module contains its own promoter as shown. EutM building block expression and secretion by *B*.

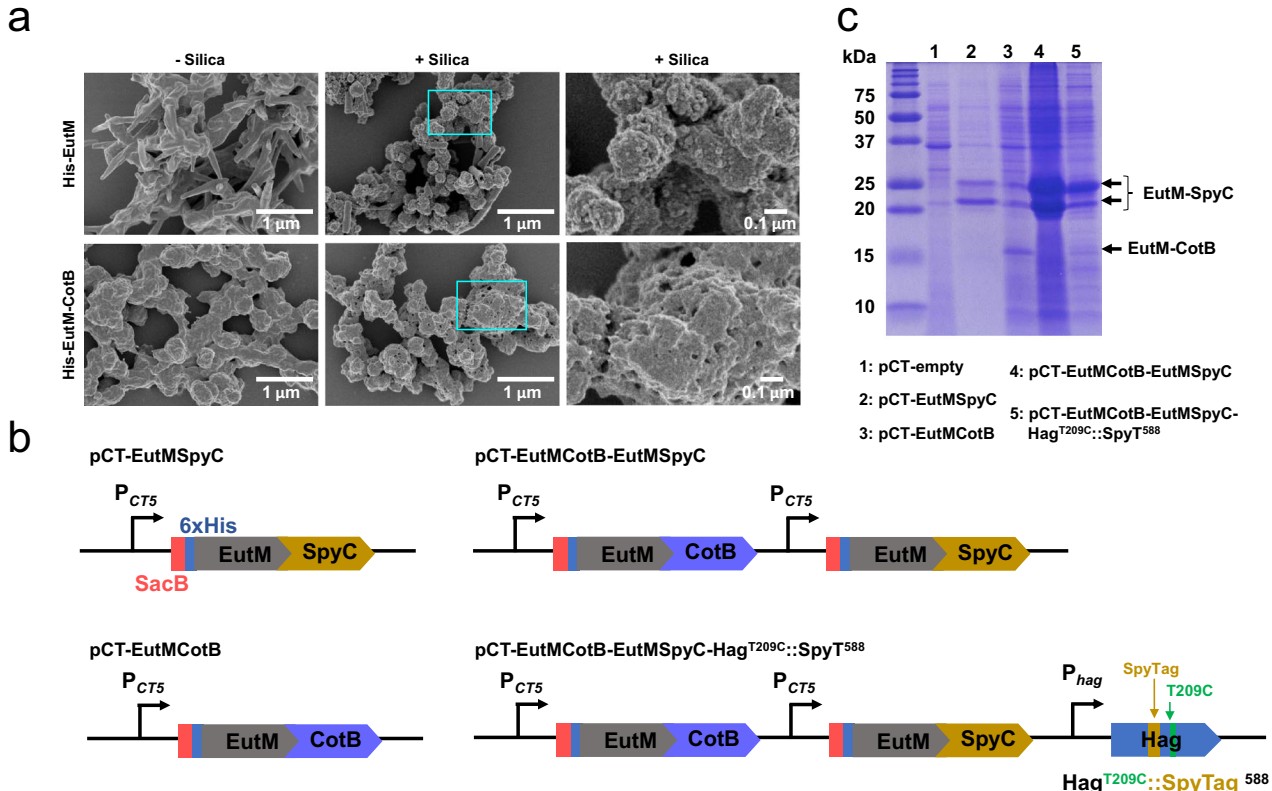

**Fig. 5 Implementing silica biomineralization by scaffolds. a** Silica biomineralization by His-EutM and His-EutM-CotB scaffolds expressed and purified from *E. coli* was investigated by SEM following the incubation of 1 mg/mL protein without (left column) and with (center columns) 100 mM silica for 2 h at room temperature. Right columns show 10-fold magnification of marked regions. Images shown are representative of one set of purified proteins. **b** Different plasmids were constructed for EutM-CotB expression and secretion in *B. subtilis* alone or together with EutM-SpyCatcher as shown. For final ELM fabrication, a consolidated plasmid was designed for the co-expression of EutM-SpyCatcher and EutM-CotB with Hag$^{T209C}$::SpyTag$^{588}$ in *B. subtilis* Δ*lyt*C Δ*flh*G. Each gene is expressed by its own promoter (cumate inducible P$_{CT5}$ promoter, native *hag* promoter). **c** Expression and secretion of EutM-CotB by *B. subtilis* Δ*lyt*C Δ*flh*G was investigated and compared to EutM-SpyCatcher. Urea solubilized fractions were prepared from cultures transformed with different expression plasmids and grown as in Fig. 2. Except for pCT-EutMSpyC (lane 2), all urea fractions from the different recombinant cultures were concentrated 10-fold by TCA precipitation prior to SDS-PAGE analysis. Data shown are representative of three independent experiments. Source data are provided as a Source Data file.

*subtilis* Δ*lyt*C Δ*flh*G transformed with the different plasmids was then confirmed and compared, showing again lower secretion levels for EutM-CotB compared to co-expressed EutM-Spy-Catcher in SMM at 20 °C after 48 h of induction (Fig. 5c). We also attempted to visualize by fluorescence microscopy the flagella phenotypes of *B. subtilis* Δ*lyt*C Δ*flh*G cultures transformed with the EutM co-expression constructs with and without Hag$^{T209C}$::-SpyTag$^{588}$. We observed large structures labeled with cysteine reactive dye that interfere with cell labeling (Supplementary Fig. 12). In contrast, cells that expressed only the modified Hag$^{T209C}$ or Hag$^{T209C}$::SpyTag$^{588}$ contained clearly labeled flagella under the same growth conditions (Supplementary Fig. 12). We therefore presume that the dye-labeled structures represent secreted EutM scaffold where one or both cysteine residues of the EutM proteins are labeled.

**Biocomposite ELM fabrication and characterization**. For the final fabrication of our biocomposite ELM we performed biomineralization experiments with induced and uninduced (control) cultures of *B. subtilis* Δ*lyt*C Δ*flh*G transformed with an empty control plasmid or one of the EutM-SpyCatcher and EutM-CotB co-expression plasmids with or without Hag$^{T209C}$::-SpyTag$^{588}$. After 48 h of cultivation, 5 mL culture samples were transferred and further incubated with and without 100 mM silica in 6-well plates at 20 °C, 100 rpm for 1 h (Fig. 6a, see

Supplementary Fig. 13 for biological replicates). While all cultures show some silica condensation into clusters at charged surfaces present in the cultures, only cultures that expressed and secreted EutM scaffolds (induced cultures) formed aggregated silica materials. Larger silica material aggregates (about twice the size) were formed in cultures that also displayed SpyTagged flagella for material cross linking.

To test whether the silica materials made by the engineered *Bacillus* strains above have different mechanical properties due to scaffold secretion and cross-linking, we optimized conditions for the formation silica blocks suitable for the rheological analysis (Fig. 6b). Rheological measurements have also been performed for other types of living materials, such as for example a cellulose-based material of a bacterial and yeast co-cultures with programmable weakening of material microstructure[27]. For measurements of our materials, induced culture samples from Fig. 6a were incubated in a syringe with 200 mM silica at 20 °C or 25 °C for up to 5 h until solid gel blocks were obtained (25 °C, 5 h) for measurements with a rheometer. Small, but statistically significant increases ($p < 0.05$) in storage moduli (G') were measured for cultures co-expressing EutM scaffold building blocks (1.23-fold for pCT-EutMCotB-EutMSpyC and 1.37-fold for pCT-EutMCotB-EutMSpyC-Hag$^{T209C}$::SpyT$^{588}$) compared to the empty plasmid control (pCT-empty) (Fig. 6c, Supplementary Table 2 and 3 for statistical analysis). The loss modulus (G")

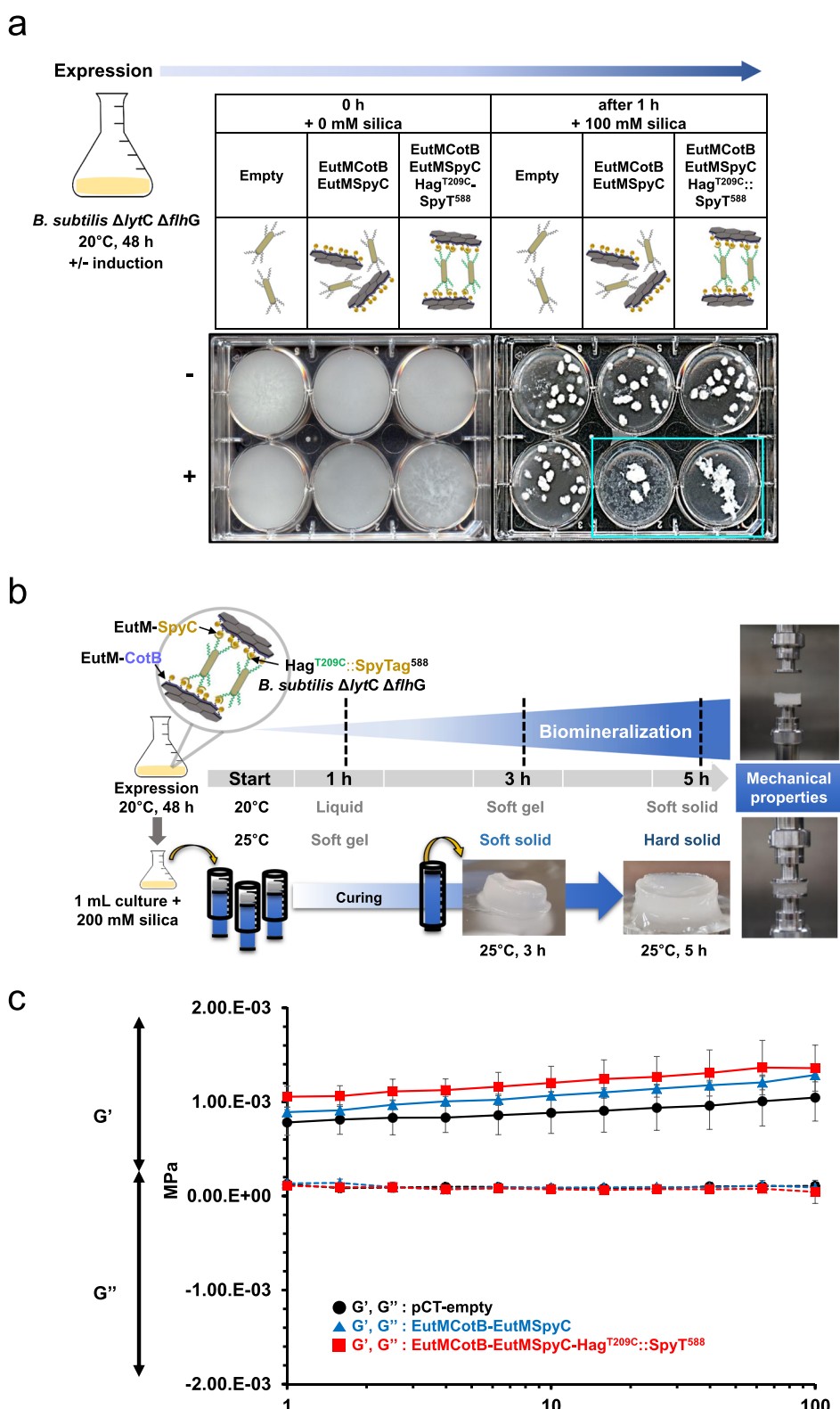

remained the same for all samples and was lower than G' as would be expected for materials with lower viscous dissipation. The increases in G' shows that the silica gel blocks from scaffold secreting cultures have a higher degree of intermolecular bonding giving them greater mechanical rigidity. The engineered *Bacillus* strain with the SpyTag displaying flagella yielded the most robust material due to cross-linking of cells and scaffolds.

Although we have shown that our final ELM *Bacillus* strain (*B. subtilis* Δ*lyt*C Δ*flh*G pCT-EutMCotB-EutMSpyC-Hag^T209C::SpyT^588) improved material strength, we wanted to determine where the secreted scaffolds are located in the material. Uninduced and induced cultures used for silica material formation in Fig. 6 were incubated with SpyTag-eGFP or eGFP (control) to label SpyCatcher domains on EutM scaffolds not linked to SpyTag

**Fig. 6 Formation of living, silica biocomposites with increased rigidity. a** Generation of silica biocomposites was tested by cultivating B. subtilis ΔlytC ΔflhG transformed with plasmids (empty as control) for expression of the shown proteins and their functions. Cultures were first grown (SMM, 20 °C, 48 h) with (+ induced) and without (- uninduced) induction of protein expression. 5 mL cultures were transferred to 6-well plates (0 h, 0 mM silica) and incubated (20 °C, 100 rpm) for 1 h after addition of 100 mM silica (1 h, 100 mM). Experiments were performed with three biological culture replicates (see Supplementary Fig. 13). **b** 3D blocks of silica biocomposite materials were then fabricated from cultures of the same three strains grown under the same conditions with induction. After 48 h of growth, 200 mM silica was added into 1 mL cultures in syringes as molds and the aliquots of the cultures were cured at 20 °C or 25°C. At different time points, silica gel plugs were removed from their molds to evaluate their hardness. After 5 h of curing at 25°C, solid gel plugs suitable for rheology testing were obtained. **c** Rheological properties of the cured gel plugs were measured using an extensional DMA Rheometer with 15 mm compression disks. A frequency sweep was performed with a gap of 4.5 mm and an oscillation strain of 1%. Three biological replicate gel blocks were measured for each strain. Variation was observed in the storage modulus (G′, solid line) and loss modulus (G″, dashed line). Data are shown as mean values ± SD and error bars represent the standard deviations of three independent biological samples. Source data are provided as a Source Data file.

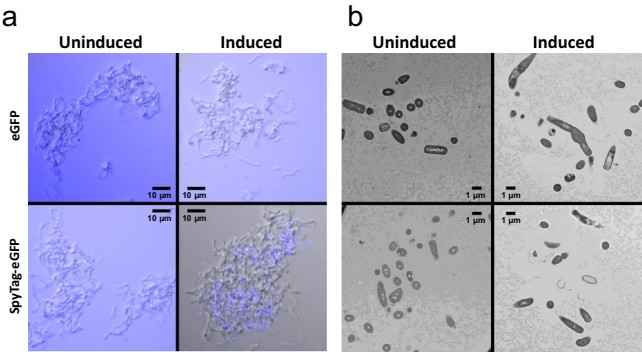

**Fig. 7 Microscopy characterization of *Bacillus* silica biocomposites. a** The location of assembled scaffolds was tested by cultivating *B. subtilis* ΔlytC ΔflhG transformed with EutMCotB-EutMSpyC-Hag^T209C::SpyT^588 plasmid (uninduced as control). Cultures were first grown (SMM, 20 °C, 48 h) with (induced) and without (uninduced) cumate induction of protein expression. 500 μL of culture was mixed with 0.125 mg eGFP or SpyTag-eGFP and mixed at RT for 30 min before acquiring DIC and GFP images which were merged and GFP channel colored blue. Induced cultures have EutM scaffolds assembled outside and between cells, with available binding sites for additional SpyTagged proteins. **b** 3D blocks of silica biocomposite materials were then formed from cultures of the same sample examined in **a**. 200 mM silica was added after 48 h of growth and 1 mL aliquots of the cultures were transferred to syringes as molds for curing at 25 °C. After 5 h of curing at 25 °C, solid gel plugs were cut into 2 mm³ pieces and prepared for thin sectioning. Cells were observed in clusters for both the uninduced and induced cultures, however only the induced culture contained clear zones around cells (see Supplementary Fig. 14a for a close-up). This zone corresponds to the SpyTag-eGFP labeled areas above and indicates there is interference with staining the scaffolds after silica is biomineralized. (Images shown are representative of two independent experiments.).

displaying flagella of cells (Fig. 7a). No labeling was observed with uninduced or eGFP control samples. In contrast, regions outside and in between cells were labeled in the induced samples, suggesting that EutM proteins are secreted and assemble into scaffolds between the cells. The relatively high binding of SpyTag-eGFP indicates that there are many available binding sites for future incorporation of additional, secreted SpyTagged cargo proteins for the fabrication of functional ELMs.

The same cultures examined by GFP labeling were then transformed into a silica blocks as in Fig. 6b for TEM visualization of cells in the biocomposite material. After curing, solid gel plugs were cut into 2 mm³ pieces and processed for thin sectioning. Cell were observed in clusters for both the induced and uninduced cultures in relatively low density which corresponds to a final OD$_{600}$ of ~3.5 (induced) or ~6 (uninduced) of the cultures prior to silica material formation (Fig. 7b, Supplementary Fig. 14a). Unexpectedly,

no scaffolds could be visualized outside or in between cells in the material from the induced cultures. Instead, only thin sections from the induced cultures contained large clear zones around cells. These zones correspond to the SpyTag-eGFP labeled areas (Fig. 7a, Supplementary Fig. 14a) and indicates that a material is physically taking up the space that does not stain after silica biomineralization. The same observations were made when the cell density in silica blocks was increased to an OD$_{600}$ of 10 and 20 by gentle centrifugation and resuspension of cells prior to biomineralization (Supplementary Fig. 14b, c). We therefore theorize that silica deposition on EutM scaffolds surrounding the cells prevents staining. Although no scaffolds were visualized in thin sections, silica gels made in the same manner above show protein bands of the expected sizes for both EutM-SpyCatcher and biomineralizing scaffolds on SDS-PAGE analysis (Supplementary Fig. 14d) Supplementary our theory that silica biomineralization is interfering with staining scaffolds during thin sectioning.

**Regeneration and functional augmentation of biocomposite ELM.** Ultimately, a living composite material for application as e.g. functional coating or plaster should be self-regenerating and integrate desired functions through their living components (Fig. 1). We therefore first tested whether our ELM could be regenerated from a small plug (~5 mm³) taken from silica blocks generated from cultures of our final ELM strain. Blocks fabricated with 200 mM silica were cured for 1 day at 25 °C and then small pieces were used to inoculate fresh cultures for expression and secretion of scaffold building blocks (Fig. 8a). After 48 h of growth with and without induction (control), secretion of EutM proteins at similar levels as the parent culture could be confirmed in the induced cultures (Fig. 8b). Similarly, we could reproduce the formation of cross-linked silica material in 6-well plates with the induced cultures (Fig. 8c). These results show that the recombinant *Bacillus* cells remain viable in the silica material and retain their engineered functions. We repeated the same experiments with silica blocks that were cured for 2 weeks at 25 °C. Although cultures could be regrown in selective media, meaning that the plasmid is still present, no scaffold building block secretion could be detected. Due to plasmid-based expression of our proteins, we theorized this issue might be caused by mutations in their coding sequences. Sequencing of plasmids isolated from individual colonies derived from the revived culture indeed confirmed that all of them contained multiple mutation in the EutM gene regions. The rapid mutability of high copy number plasmids in *Bacillus* is a well-known issue[68,69] and can be solved in the future by integrating the expression cassettes currently located on a plasmid into the genome of our engineered *Bacillus* strain. Nevertheless, the ability to regrow the engineered strain with its plasmid after 2 weeks in a silica gel speaks to the resilience of *B. subtilis* as ELM chassis organism.

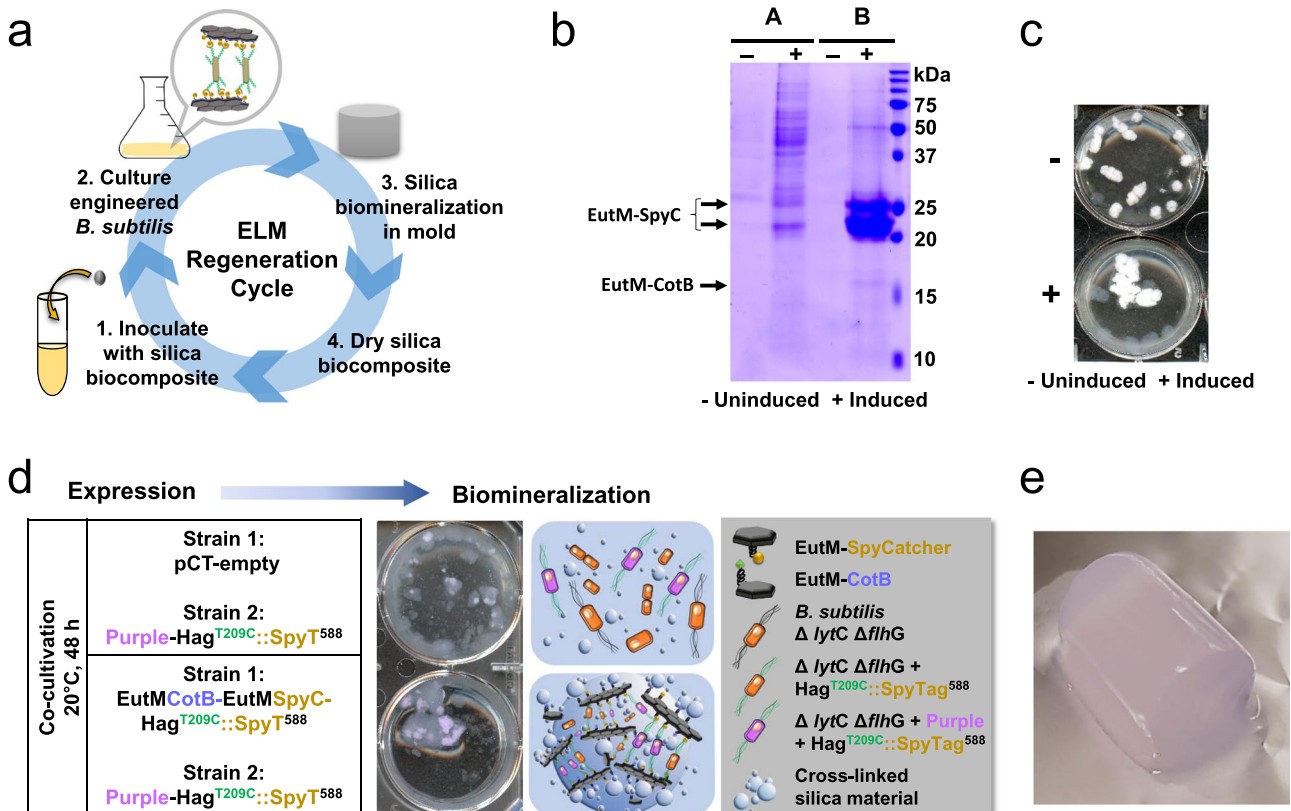

**Fig. 8 ELM biocomposite regeneration and proof-of-concept for material functionalization. a** Silica biocomposite material regeneration was tested by re-inoculating cultures with a piece of a silica plug fabricated from *B. subtilis* Δ*lyt*C Δ*flh*G cultures co-expressing EutM-CotB and EutM-SpyCatcher and Hag[T209C]SpyTag[588], biomineralized with 200 mM silica (see Fig. 6) and cured for 24 h at 25 °C. Cultures were regrown (SMM, 20 °C, 48 h) with (+ induced) and without (- uninduced) cumate induction of protein expression. **b** Expression and secretion of EutM scaffold building blocks by the induced, regenerated cultures was confirmed by SDS-PAGE analysis of culture supernatant (A) and urea solubilized scaffolds (B). All fractions were concentrated 10-fold by TCA precipitation prior to SDS-PAGE analysis. **c** Silica biomineralization into a cross-linked biocomposite material was confirmed for the induced, regenerated cultures after 1 h at 20 °C with 100 mM silica in 6-well plates as in Fig. 6. **d** Proof-of-concept incorporation of additional functions into the silica ELM was demonstrated by the co-cultivation of two strains, one strain co-expresses scaffold building blocks and SpyTagged flagellin, while a second strain co-expresses a purple chromophore protein and SpyTagged flagellin. In the control co-culture, the plasmid for scaffold building block secretion in one strain was replaces with an empty plasmid. Strains were inoculated at a 1:1 ratio and cultures grown under inducing conditions for biomineralization as described above. Only co-cultures that co-expressed EutM-CotB and EutM-SpyCatcher formed an aggregated silica material with embedded purple cells. **e** A block of a purple, silica biocomposite (15 mm × 30 mm) was then fabricated from co-cultures of engineered strains co-expressing scaffold building blocks and the purple protein together with SpyTagged flagellin. A solid material was formed with 200 mM silica after curing for 24 h at 25 °C. All results shown are representative of three independent biological samples. Source data are provided as a Source Data file.

As a proof-of-concept, we sought to demonstrate the incorporation of another property into our material. Rather than adding another genetic burden to our engineered *Bacillus* strain, we opted to co-cultivate our engineered strain with a second engineered *Bacillus* strain that would provide the new property. Such a co-cultivation strategy but with two different microorganisms has recently been successfully explored for the fabrication of a living material that incorporates cellulose[27]. For demonstration purposes, we chose the formation of colored material by expressing a purple chromoprotein[70]. An additional plasmid pCT-Purple-Hag[T209C]::SpyT[588] was constructed for transformation into *B. subtilis* Δ*lyt*C Δ*flh*G and co-cultivation with our final ELM strain. Co-cultures (mixed at a 1:1 ratio in the induction step) were grown as before (Fig. 6) and biomineralization and silica material formation compared to co-cultures with an empty plasmid control (Fig. 8d). Only co-cultures that also expressed the EutM scaffolds formed a purple a cross-linked silica material due to the attachment of the purple protein-expressing *Bacillus* cells (which also display SpyTags on their flagella) to the scaffolds. Likewise, a purple-colored silica block could be fabricated under the same conditions used above with monocultures of our

engineered *Bacillus* strain (Fig. 8e). These results demonstrate an easy and modular approach for the fabrication of living materials with tailored functions and properties ranging e.g., from color and antimicrobial properties[71] to more complex functions, such as sensing and responding.

We successfully engineered *B. subtilis* to produce and secrete proteins that self-assemble into scaffolds capable of biomineralizing silica, forming a material with tailorable properties and functions via its ability to crosslink cells and covalent attachment of cargo proteins. Our ELM system does not rely upon special processing after cultivation and instead just requires the addition of silica. Microscopy and rheological data show this ELM system function effectively to secrete biomineralizing and cross-linking scaffold that produce a material with stronger mechanical properties than the nonscaffold secreting controls. Although silica gel formation requires a concentration of 100 mM silica (derived from hydrolyzed TEOS), which is currently much higher than used by natural systems which can biomineralize silica through concentration mechanisms from less than 100 μM available silica in their environments[72], the use of hydrolyzed TEOS as a source of silica is inexpensive. Additional research to

enhance biomineralization by engineering and increasing the density of biomineralization peptides on scaffolds could reduce silica concentrations, and the incorporation of different types of biomineralization peptides[60–64] would give access to a range of other ELM biocomposite materials. We have shown that our ELM can be regenerated from a piece of silica material containing cells after 24 h of curing. But extended material storage and regeneration attempts resulted in the accumulation of mutations in plasmids recovered from regrown cells. We utilized plasmids for easy optimization of protein expression, but these expression constructs once designed in their final configurations will need to be integrated into the genome to ensure stable, genetic programming of ELMs that can regenerate over many cycles. Finally, this work offers engineering strategies for expanding the repertoire of current ELM chassis organisms towards other spore-forming and/or ultra-resilient bacteria that secrete and embed themselves into a genetically programmable protein matrix for the fabrication of a range of functional materials.

## Methods

**Materials and chemicals**. Reagent grade tetraethyl orthosilicate (TEOS), ammonium molybdate tetrahydrate, 4-methylaminophenol sulfate, anhydrous sodium sulfite, all acids and bases, and other chemicals were purchased from Sigma-Aldrich (Sigma-Aldrich Corp., MO, USA).

Remel RYU flagella stain, polysine microscope slides, plain precleaned glass microscope slides, Alexa Fluor™ 488 C₅ maleimide, Syto16, and Trump's Fixative were purchased from ThermoFisher Scientific (Waltham, MA, USA). 22 mm 1.5 cover glasses were purchased from Azer Scientific (Morgantown, PA, USA). 200 mesh copper grids with 10 nm formvar and 1 nm carbon, aqueous 2% uranyl acetate, and EM grade: 10% glutaraldehyde, 0.2 M sodium cacodylate trihydrate pH 7.4, 16% paraformaldehyde, and 4% osmium tetraoxide were purchased from VWR International, an Electron Microscopy Sciences reseller (Radnor, Pennsylvania, USA). SynaptoRed™ C2 was purchased from Avantor delivered by VWR a Biotium reseller (Radnor, PA, USA). FluoroShield was used in fluorescent microscopy experiments and was purchased from AbCam (Cambridge, MA, USA). Silicon wafers were used for SEM and obtained from Ted Pella (Reeding, CA, USA). Schaeffer and Fulton Spore stain kit (containing malachite green 50 g/L and safranin O 5 g/L) was obtained from MilliporeSigma (Burlington, MA, USA).

Spectra/Por Dialysis Tubing (MWCO 6-8 kDa) used for dialysis of proteins was purchased from Spectrum LifeSciences (Rancho Dominguez, CA, USA). HisTrap™ FF columns (GE Healthcare, IL, USA) were used for all protein purifications except for proteins used for TEM that were purified from *E. coli* and secreted proteins from *Bacillus* that were purified using TALON metal affinity resin from TaKaRa Bio USA, Inc. (Mountain View, CA, USA). Two mL of TALON metal affinity resin was used in a Batch/Gravity flow protein purification protocol following manufacturer's directions. Pierce™ BCA Protein Assay Kit used for protein concentration measurements was purchased from ThermoFisher Scientific (Waltham, MA, USA). Ultrapure water was prepared by filtering deionized water through a Milli-Q water purification system (Millipore, Billerica, MA, USA) to reach a final electrical resistance higher than 18.2 MΩ cm⁻¹.

All supplies were purchased from New England Biolabs (NEB) (Ipswhich, MA, USA). Q5® High-Fidelity DNA polymerase was used for PCR amplifications. The Q5® Site-Directed Mutagenesis kit was used for site-directed mutagenesis and the NEBuilder® HiFi DNA Assembly Master Mix was used for Gibson assembly of DNA fragments, and restriction enzymes (*Bgl*II, *Bam*HI, and *Pst*I) and T4 DNA Ligase were used for general cloning. Primers were ordered from Integrated DNA Technologies (Coralville, IA, USA).

40% acrylamide and bis-acrylamide solution (37.5:1), TEMED and Precision plus all blue prestained protein standard (catalog# 161-0373) were purchased from Bio-Rad (Hercules, CA, USA). Other chemicals for SDS-PAGE analysis and Coomassie brilliant blue staining were purchased from Sigma-Aldrich.

**Bacterial strains, media, and general molecular biology methods**. *Escherichia coli* TOP10 (Invitrogen, Carlsbad, CA, USA) was used for cloning and plasmid propagation. *E. coli* T7 Express (C2566) (NEB) and used for protein expression and purification. *Bacillus subtilis* 168 (ATCC 23857) was used as a host strain (referred to as WT strain). All plasmids and strains used in this study are listed in Supplementary Data 1.

LB (Luria broth; Tryptone 10 g/L, NaCl 5 g/L, Yeast extract 5 g/L) medium was used for the growth of *E. coli* and *B. subtilis* strains.

*B. subtilis* strains were also grown in modified Spizizen's minimal medium (SMM)[73,74], containing, per liter: K₂HPO₄–17.5 g, KH₂PO₄–7.5 g, Na₃-Citrate 2 H₂O–1.25 g, MgSO₄ 7 H₂O - 0.25 g, N-source (L-glutamine–10 g, (NH₄)₂ SO₄–0.5 g), trace elements (CaCl₂ 2 H₂O - 7.35 mg, MnCl₂ 4 H₂O–1 mg, ZnCl₂–1.7 mg, CuCl₂ 6 H₂O–0.43 mg, CoCl₂ 6 H₂O–0.6 mg, Na₂MoO₄ 2

H₂O–0.6 mg), Fe-Citrate (FeCl₃ 6 H₂O–1.35 mg, Na₃-Citrate 3 H₂O–10 mg), glucose–5 g, tryptophan–50 mg.

SMM was prepared by mixing stock solutions which included, 5X SMM base (K₂HPO₄–87.5 g/L, KH₂PO₄–37.5 g/L, Na₃-Citrate·2H₂O–6.25 g/L, MgSO₄ · 7H₂O–1.25 g/L), N-source (sterilized using a 0.2 µm filter) (glutamine–40 g/L, (NH₄)₂SO₄–2 g/L), 100X trace elements (CaCl₂ · 2H₂O–0.735 g/L, MnCl₂ · 4H₂O–0.1 g/L, ZnCl₂–0.17 g/L, CuCl₂ · 6H₂O–0.043 g/L, CoCl₂ · 6H₂O–0.06 g/L, Na₂MoO₄ · 2H₂O–0.06 g/L), 100X Fe-citrate (sterilized using a 0.2 µm filter) (FeCl₃ 6 H₂O–0.135 g/L, Na₃-Citrate.3H₂O–1 g/L), 50% glucose (500 g/L) and tryptophan (sterilized using a 0.2 µm filter) (5 g/L).

Fresh solution of sterile N-source and glucose were prepared every time while stocks of other solutions were stored at 4 °C. The working solution of SMM was prepared by mixing 200 mL of 5X SMM, 10 mL of 100X trace element, 10 mL of 100X Fe-Citrate, 10 mL of 50% glucose, 10 mL from 5 mg/mL stock of Tryptophan, and 250 mL of N-source into 510 mL of sterile water.

SM1 and SM2 media as described previously by Bennallack et al.,[75] were used for the preparation of *B. subtilis* competent cells.

For plasmid maintenance, LB was supplemented with 100 µg/mL ampicillin for *E. coli* strains while LB and SMM were supplemented with 10 µg/mL tetracycline or 20 µg/mL erythromycin for *Bacillus* strains.

Transformation of *E. coli* with plasmids followed standard molecular biology techniques. Transformation of *B. subtilis* strains was done using a method described by Bennallack et al.[75] Transformants were screened by colony PCR using a method modified from Jamal et al.[76] To prepare samples for colony PCR, a single colony of *B. subtilis* was mixed in 300 µL sterile water. Cells were spun down (21,130 x *g*) and 200 µL of the supernatant was removed. Cells were resuspended in the remaining 100 µL of sterile water followed by 20 mins of sonication in a coldwater bath sonicator (Bransonic 3510R-MTH, CT, USA). After sonication, cells were spun down as above and 2 µL of the supernatant was used as a template for a 10 µL PCR reaction. Plasmids from positive *B. subtilis* transformants were isolated and their sequences verified by Sanger sequencing. Recombinant strains with verified plasmid sequences were stored as glycerol stocks and used in this work.

**Construction of plasmids and in-frame deletion in *B. subtilis***. All plasmids and strains used in this study are described in Supplementary Data 1. Plasmid sequences were verified by Sanger sequencing (ACGT, Inc. Wheeling, IL, USA). Amino acid sequences and nucleotide sequences for all proteins and genes reported in this work are found in Supplementary Data 2 and 3. Primer pairs for *Bacillus* genomic manipulations and flagellin engineering are listed in Supplementary Data 4.

Plasmids were constructed using a combination of Gibson Assembly (HiFi® DNA assembly kit, NEB) for fragment assembly, site-directed mutagenesis (Q5® mutagenesis kit, NEB) for shorter insertions, deletions, and insertions, or restriction enzyme cloning and ligation for larger fragment assembly following manufacturers' instructions and as described previously[33,41].

Briefly, for the C-terminal fusion of biomineralization peptides to His-EutM in pCT5-His-EutM[33,41], primers were designed to insert the corresponding into the plasmid using the Q5® mutagenesis kit according to the manufacturers' instructions.

To construct the pCT-His-dtTomato-SpyCatcher plasmid, His-EutM was combined with the fluorescent protein in a previously constructed pCT5-His-EutMSpyC plasmid[35] by HiFi Assembly. His-tdTomato-SpyCatcher was amplified from tdTomato-pBAD[77].

To create the EutM *Bacillus* expression vectors, His-EutM-SpyCatcher was amplified from pCT5-EutMSpyC and inserted into our in-house, cumate inducible *Bacillus* expression vector pCT5-bac2.0[41] by HiFi Assembly. For a control plasmid, pCT-empty was created by deletion of the *sfgfp* gene from pCT5-bac2.0 using the Q5® mutagenesis kit according to the manufacturers' instructions. All signal peptides (Supplementary Table 1) were fused to His-EutM-SpyCatcher by site-directed mutagenesis (Q5® mutagenesis kit, NEB) with appropriate primers. The pCT-EutMCotB plasmid for *Bacillus* expression was made by replacing the SpyCatcher sequence in pCT-EutMSpyC with the CotB sequence by site-directed mutagenesis. To create pCT-EutMCotB-EutMSpyC, the complete SacB-His-EutM-SpyCatcher expression cassette (including cumate inducible promoter) from pCT-EutMSpyC was amplified and subcloned into pCT-EutMCotB. The hag^T209C::SpyTag^588 expression cassette (including its native promoter) was added to pCT-EutMCotB-EutMSpyC by HiFi assembly with the hag expression module amplified from the corresponding pBBRm34 plasmid (see below). Finally, to construct pCT-Purple-Hag^T209C::SpyT^588, the chromoprotein was obtained from pSB1C3-tsPurple[70] and assembled into pCT-EutMCotB-EutMSpyC-hag^T209C::SpyT^588 by replacing the EutM expression modules.

The flagellin gene (*hag* gene) was amplified from *B. subtilis* genomic DNA using primer pair (NP-hag_fwd and NP-hag_rev) and cloned into the pRBBm34 backbone using HiFi assembly. The flagellin T209C mutation was introduced by Q5®-mutagenesis with primer pairs (NP-hag-T209C_fwd and NP-hag-T209C_rev). A SpyTag was inserted into nine different sites between D1b-N and D1-C domains using Q5®-mutagenesis with primer pairs listed in Supplementary Data 4.

To generate the Δ*lyt*C marker-less deletion construct, the pHBintE vector carrying a temperature-sensitive origin of replication, erythromycin and ampicillin resistance was used as the backbone. The *lyt*C upstream region was amplified from *B. subtilis* genomic DNA using primer pair d*lyt*C_upstream_fwd and d*lyt*C_upstream_rev, while the *lyt*C downstream region was amplified using primer pair d*lyt*C_downstream_fwd and d*lyt*C_downstream_rev (Supplementary Data 4) and the deletion cassette was cloned into pHBintE vector using HiFi assembly. The plasmid for Δ*flh*G marker-less deletion was created in a similar manner: The *flh*G upstream region was amplified using primer pair d*flh*G_upstream_fwd and d*flh*G_upstream_rev while the *flh*G downstream region was amplified using primer pair d*flh*G _downstream_fwd and d*flh*G_downstream_rev (Supplementary Data 4). The *flh*G deletion cassette was cloned into pHBintE vector using HiFi assembly.

The *lyt*C deletion plasmid (temperature-sensitive pHBintE-d*lyt*C) was transformed into *B. subtilis* using the method described by Bennallack et al.[75]. After transformation, plates were incubated at a temperature permissive for plasmid replication (30 °C) and for selection of erythromycin-resistant transformants. Colonies were then transferred into 4 mL LB medium without antibiotic added followed by several rounds of dilution and regrowth at a temperature nonpermissive for plasmid replication (37 °C). For screening of the deletion mutant appropriate dilutions of the cells were plated onto LB agar. Colonies were streaked on LB plates with and without erythromycin to identify erythromycin sensitive colonies that have successfully evicted the plasmid. Screening the erythromycin sensitive colonies for desired double homologous recombinants was performed using colony PCR with primer pair d*lyt*C_confirmation_fwd and d*lyt*C_confirmation_rev as shown in Supplementary Data 4. Transformation of *flh*G deletion construct was done into *B. subtilis* Δ*lyt*C host using the pHBintE-d*flh*G plasmid. In-frame deletion of *flh*G followed by screening was performed with the same method as above for *lyt*C but the primer pair used for screening of colonies for *flh*G deletion mutants was d*flh*G_confirmation_fwd and d*flh*G_confirmation_rev.

**Cultivation of recombinant *B. subtilis* strains.** Glycerol stocks of recombinant *Bacillus* strains (WT and Δ*lyt*C Δ*flh*G deletion strain) transformed with pCT expression plasmids (pCT-empty, pCT-EutMSpyC, pCT-EutMCotB, pCT-EutMCotB-EutMSpyC, pCT-EutMCotB-EutMSpyC-Hag^T209C^::SpyT^588^) were streaked onto LB plates supplemented with tetracycline (10 μg/mL) and grown overnight at 30 °C. A single colony was used to inoculate 4 mL of selective LB. After 19 h of growth at 30 °C, 220 rpm strains were diluted 3% into 50 mL of LB or SMM supplemented with tetracycline (10 μg/mL) in a 250 mL flask. The flasks were cultured at 37 °C, 180 rpm until $OD_{600}$ reached 0.4–0.7 and protein expression was induced with 10 μM cumate. The induced cultures were then grown at 20 °C or 30 °C, 100 rpm for 48 to 96 h as described in this study. Final scaffold building block expression and secretion was done in SMM with 10 μM cumate for induction and subsequent growth at 20 °C, 100 rpm for 48 h (referred to as 'standard conditions').

For co-cultivation experiments, glycerol stocks of three *B. subtilis* Δ*lyt*C Δ*flh*G strains transformed with pCT-empty, pCT-EutMCotB-EutMSpyC-Hag^T209C^::SpyT^588^ or pCT-Purple- Hag^T209C^::SpyT^588^ were first spread onto selective LB plates, and single colonies were used to inoculate 4 mL of selective LB. After 19 h of growth at 30 °C, cultures were diluted 3% into 50 mL of selective SMM in a 250 mL flask. Individual cultures were then grown at 37 °C until $OD_{600}$ 0.4–0.7. 25 mL of each culture was then mixed in a new 250 mL flask as follows: Co-culture 1: *B. subtilis* Δ*lyt*C Δ*flh*G pCT-empty + *B. subtilis* Δ*lyt*C Δ*flh*G pCT-Purple-Hag^T209C^::SpyT^588^, Co-culture 2: *B. subtilis* Δ*lyt*C Δ*flh*G pCT-EutMCotB-EutMSpyC-Hag^T209C^::SpyT^588^ + *B. subtilis* Δ*lyt*C Δ*flh*G pCT-Purple-Hag^T209C^::SpyT^588^. Protein expression was induced with 10 μM cumate and co-cultures grown at 20 °C, 100 rpm for 48 h.

All expression and cultivation experiments were performed with three biological replicates (i.e. three independent cultures).

**Analysis of scaffold building block expression and secretion by *Bacillus* strains.** Extracellular secretion and intracellular accumulation of scaffold building blocks by recombinant *Bacillus* strains was analyzed by preparing four different fractions from cultures for SDS-PAGE analysis: (i) culture supernatant, (ii) urea supernatant containing urea solubilized scaffolds that co-precipitated with cells, (iii) lysed cell pellets after solubilization of co-precipitated scaffolds, and (iv) total protein (secreted-precipitated and intracellular) from completely lysed cells with sonication followed by lysozyme treatment. The preparation of all four protein fractions is shown as a flow chart in Supplementary Fig. 5a. Fractions i-iii correspond to samples A, B, and C in Figs. 2, 3, and 8. Comparison of protein bands in fraction ii (i.e., absence of bands except for solubilized EutM protein) and iv indicates that urea does not break open cells Supplementary Fig. 5b.

The following workflow was used for sample preparation: 5 mL of culture broth was centrifuged at 3220 x *g*, at 4 °C for 10 mins to separate culture supernatant from pelleted cells and co-precipitated scaffolds.

For SDS-PAGE analysis of soluble scaffolds in the culture supernatant (fraction i, or A in Fig. 2), proteins from 1 mL of supernatant were precipitated by adding 200 μL of 100% Trichloroacetic acid (TCA) and 1 h incubation on ice. Precipitated proteins were centrifuged at 21,130 x *g* for 10 mins at 4 °C and the supernatant was removed. The precipitate was washed with 1 mL of pre-chilled acetone and the tube

was centrifuged again at 21,130 x *g* for 10 mins at 4 °C. The acetone was removed and the washing step was repeated twice. The dried precipitate was resuspended in 100 μL of 1X SDS-loading buffer (2% w/v SDS, 0.1 M DTT) (6X standard SDS-loading buffer diluted with 50 mM Tris-HCl, 4 M urea, pH 7.5) for a 10-fold concentrated sample.

For SDS-PAGE analysis of secreted scaffolds that co-precipitate with cells after centrifugation of 5 mL cultures (fraction ii), the collected pellet was carefully resuspended in 1 mL of urea buffer (4 M urea, 50 mM Tris-HCl, pH 7.5). The resuspended solution was centrifuged at 3,220 x *g* at 4 °C for 10 mins and the supernatant was analyzed by SDS-PAGE.

For SDS-PAGE of the remaining scaffolds in the urea washed pellet (fraction iii), cells were lysed with 200 μg/mL final concentration of lysozyme in 500 μL of 50 mM Tris-HCl, pH 7.5 with incubation at 37 °C for 1 h. The clarified supernatant after centrifugation (3,220 x *g*, 10 mins) was analyzed by SDS-PAGE.

For SDS-PAGE of total protein (fraction iv), pelleted cells and co-precipitated scaffolds from 5 mL of culture was resuspended into 500 μL of 50 mM Tris-HCl pH 7.5 in an eppendorf tube. The suspension was sonicated in a coldwater bath sonicator (Bransonic 3510R-MTH, CT, USA) followed by lysis with lysozyme as above. The resulting suspension was mixed with SDS-loading buffer.

All protein samples were denatured by boiling for 10 mins at 100 °C and 8 μL of each sample was loaded onto a 15% SDS-PAGE gel. A PowerPAC 300 power supply (Bio-Rad) was used for SDS-PAGE analysis. Coomassie brilliant blue staining was then used for visualization of protein bands.

***De novo* sequencing of secreted EutM-SpyCatcher proteins**. For *de novo* sequencing of secreted EutM-SpyCatcher, protein bands separated by SDS-PAGE were cut from gels and submitted to the Center for Mass Spectrometry & Proteomics, University of Minnesota for analysis by LC-MS using a Thermo Scientific LTQ Orbitrap Velos mass spectrometer. Data were analyzed and viewed using the PEAKS Xpro Studio 10.6 software (Bioinformatics Solutions Inc., Waterloo, ON, Canada).

**Analysis of EutM-SpyCatcher attachment to Hag^T209C^::SpyTag^588^ displaying flagella**. Covalent isopeptide bond formation between secreted EutM-SpyCatcher and Hag^T209C^::SpyTag^588^ displayed on flagella of *B. subtilis* Δ*lyt*C Δ*flh*G was analyzed by SDS-PAGE of sheared flagella isolated from co-cultures. Glycerol stocks of three *B. subtilis* Δ*lyt*C Δ*flh*G strains (pCT-empty, pCT-EutMSpyC, pRBBm34-Hag^T209C^::SpyT^588^) were first spread onto selective LB plates, and single colonies were used to inoculate 4 mL of selective LB liquid cultures. After 19 h of growth at 30 °C, cultures were diluted 3% into 50 mL of fresh, selective SMM and grown at 37 °C until $OD_{600}$ of 0.4–0.7 was reached. 25 mL of each culture was then mixed in a new 250 mL flask as follows: Co-culture 1: *B. subtilis* Δ*lyt*C Δ*flh*G pCT-empty + *B. subtilis* Δ*lyt*C Δ*flh*G pRBBm34-Hag^T209C^::SpyT^588^, Co-culture 2: *B. subtilis* Δ*lyt*C Δ*flh*G pCT-EutMSpyC + *B. subtilis* Δ*lyt*C Δ*flh*G pRBBm34-Hag^T209C^::SpyT^588^. Protein expression was induced with 10 μM cumate and co-cultures grown at 20 °C, 100 rpm for 96 h.

Cell pellets were collected from 50 mL cultures by centrifugation (3220 x *g*, 30 mins) and resuspended in 20 mL PBS (pH 7.5). The suspension was then sonicated at low power (30% power, 5 sec on and 5 sec off, three times) (Branson 450 Digital Sonifier, CT, USA) to shear off flagella. Turbo^TM^ DNase (Invitrogen, Waltham, MA, USA) was then added at a 0.5 U/mL final concentration and incubated on ice for 10 mins. Cells were then removed by centrifugation at 3220 x *g* for 10 mins at 4 °C, followed by an ultra-centrifugation step (39,800 x *g* for 2 h at 4 °C) to collect sheared flagella which were then resuspended in 100 μL 50 mM Tris-HCl (pH 7.5) for SDS-PAGE analysis.

**Protein purification**. His-EutM, His-EutM-silica binding peptide fusion proteins and His-EutM-SpyCatcher were expressed in *E. coli* and purified by metal affinity chromatography as described in detail in:[33–35,40]. Briefly, proteins were isolated from 500 mL recombinant *E. coli* C3566 LB cultures (37 °C, 220 rpm) transformed with pCT5-plasmids encoding EutM proteins (Supplementary Data 1). Gene expression was induced with 50 μM cumate at $OD_{600}$ of 0.4–0.7 and cultures continued to be incubated overnight. Cells were collected by centrifugation (4000 x *g*, 30 min, 4 °C) and resuspended into either 30 mL EutM purification buffer (20 mM Tris-HCl, 250 mM NaCl, 5 mM imidazole, 4 M Urea, pH 7.5) for His-EutM and His-EutM-silica binding peptide fusion proteins, or our EutM-SpyCatcher purification buffer (50 mM Tris-HCl, 250 mM NaCl, 20 mM imidazole, pH 8.0)[35]. Resuspended cells were lysed by sonication on ice (30 min, power 50%, pulse on 10 s, pulse off 20 s with a Branson Sonifier) followed by centrifugation (12,000 x *g*, 40 min, 4 °C) and passing of the supernatant through 0.2 μm ultrafilter. The soluble protein was then loaded onto a 5-mL HisTrap^TM^ HP column (GE Healthcare Life Sciences, Pittsburgh, PA) using an ÄKTA FPLC system. Bound protein was eluted with 250 mM imidazole added to the purification buffer. His-tdTomato-SpyCatcher protein was expressed *a*nd purified using the EutM-SpyCatcher protocol[35].

For silica binding assays and SEM analysis of purified His-EutM and His-EutM silica binding fusion protein, purified proteins were dialyzed against deionized water.

For TEM analysis of EutM-SpyCatcher scaffold formation in SMM, proteins expressed in *E. coli* or secreted by *Bacillus* cultures were purified with a 2 mL TALON metal affinity resin using a Batch/Gravity flow protein purification protocol following manufacturer's directions (TaKaRa Bio USA, Inc., see above). EutM-SpyCatcher protein from *Bacillus* was eluted with the same elution buffer (50 mM Tris-HCl, 250 mM NaCl, 250 mM imidazole pH 8.0) used for His-EutM-SpyCatcher as previously described[33–35,40]. EutM-SpyCatcher from *E. coli* was isolated from 125 mL cultures using the EutM-SpyCatcher protocol[35] but with a TALON gravity column instead of a HisTrap™ FF column (GE Healthcare, IL, USA). For the purification of His-EutM-SpyCatcher proteins secreted by *B. subtilis* Δ*lyt*C Δ*flh*G, recombinant strains harboring pCT-EutMSpyC were grown and induced under standard conditions described above. Cells and scaffolds from twelve 50 mL cultures (total volume: 600 mL) were pelleted at 3,220 x *g*, at 4 °C for 10 mins and scaffolds solubilized into 30 mL of 50 mM Tris-HCl, pH 7.5 containing 4 M urea as described above. The solubilized scaffolds were then purified by TALON metal affinity chromatography as described above for *E. coli*. Purified protein from *E. coli* or *Bacillus* cultures were then dialyzed against SMM after elution from the TALON metal affinity resin.

For GFP labeling of scaffolds in *B. subtilis* Δ*lyt*C Δ*flh*G transformed with EutMCotB-EutMSpyC-Hag[T209C]::SpyT[588] plasmid, His-tagged eGFP and SpyTag-eGFP were expressed in *E. coli* from pET28 expression plasmids and then purified by metal affinity chromatography as described in[35].

**Preparation of hydrolyzed TEOS as silica source**. Silicic acid solutions (i.e. silica) were prepared by hydrolyzing TEOS. TEOS was hydrolyzed by adding water and 1 M hydrochloric acid at a 1:5.3:0.0013 molar ratio of TEOS:water:HCl and stirring vigorously. Under these conditions, hydrolysis is complete after 80 mins[78] for use as silica source in silica binding and biomineralization experiments.

**Silica binding assays on EutM scaffolds**. Silica precipitation was measured using the blue silicomolybdate method[66,79]. Silica was added to purified protein scaffolds, at a concentration of 1 mg/mL in deionized water, to a concentration of 100 mM and the suspension was mixed for 2 h at room temperature (RT). The precipitate was then collected by centrifugation (3 mins at 18,800 x *g*), and the resulting pellet was washed 3 times with deionized water. Precipitated silica in the scaffold pellet was depolymerized by adding 1 M NaOH for 10 mins. The samples were then diluted to 1.6 mL and 0.15 mL of solution A (containing 20 g/L ammonium molybdate tetrahydrate and 60 mL/L concentrated hydrochloric acid in deionized water) was added to the sample. After 10 mins, 0.75 mL of solution B (containing 20 g/L oxalic acid, 6.67 g/L 4-methylaminophenol sulfate, 4 g/L anhydrous sodium sulfite, and 100 mL/L concentrated sulfuric acid in deionized water) was added. The blue color was allowed to develop for 2 h at RT, and the absorbance was measured at 810 nm. All samples were run in triplicate.

**Optimization of silica concentration for EutM-CotB mediated biomineralization**. To test and optimize conditions for silica gel formation by secreted EutM-CotB scaffolds, recombinant *Bacillus* cultures were incubated with six different concentrations of silica. *B. subtilis* WT strain harboring pCT-EutMCotB or pCT-empty (control) were first cultured in 50 mL SMM under optimal scaffold secretion conditions described above (20 °C, 100 rpm, for 48 h with 10 μM cumate added for induction of protein expression). Cultures were grown as three independent biological replicates. $OD_{600}$ and pH measured after 48 h were for *B. subtilis* pCT-empty: $OD_{600} = 1.73 \pm 0.09$, pH = $6.94 \pm 0.01$, and for *B. subtilis* pCT-EutMCotB: $OD_{600} = 1.26 \pm 0.16$, pH = $6.90 \pm 0.02$. Then, 2.5 mL culture aliquots were transferred to fresh culture glass tubes, incubated with 50–500 mM silica at 20 °C for 24 h and finally inspected for gel formation.

**Six-well plate biomineralization experiments**. Five mL of *Bacillus* cultures were transferred into sterile 6-well plates. Silica was added to cultures to a concentration of 100 mM. Samples were allowed to gel at 20 °C, 100 rpm for 1 h. The biomineralization experiments were performed with three biological replicates (i.e., three independent cultures for each experiment).

**Silica block formation and mechanical testing**. Silica gel plugs were prepared in 10 mL syringes (ID = 15 mm) with the tops removed and then tops were covered instead with sterilized caps to maintain sterile conditions. Silica was added to 1 mL cultures to a concentration of 200 mM. Samples were incubated at 25 °C for 5 h, and the gel plugs were pushed out of the syringe. Mechanical properties of the gel plugs were measured using an extensional DMA Rheometer (TA Instruments RSA-G2) with 15 mm compression disks. A frequency sweep was performed with a gap of 4.5 mm and an oscillation strain of 1%.

Larger silica gel blocks from 'purple' co-cultures of *B. subtilis* Δ*lyt*C Δ*flh*G pCT-EutMCotB-EutMSpyC-Hag[T209C]::SpyT[588] + *B. subtilis* Δ*lyt*C Δ*flh*G pCT-Purple-Hag[T209C]::SpyT[588] were prepared in the same manner, except that 3 mL of culture was used and the material was cured for 24 h.

**ELM regeneration**. Small piece (~5 mm³) from silica plugs fabricated from *B. subtilis* Δ*lyt*C Δ*flh*G pCT-EutMCotB-EutMSpyC-Hag[T209C]::SpyT[588] cultures

(biomineralized with 200 mM silica and cured for 24 h at 25 °C) was used to inoculate fresh cultures for ELM fabrication under standard conditions (see above). Scaffold building block expression, secretion, and biomineralization in six-well plates by the regrown cultures was confirmed as described above.

**Protein analysis of silica gels**. Silica gel plugs (15 mm × 5 mm) from 1 mL cultures of *B. subtilis* Δ*lyt*C Δ*flh*G pCT-EutMCotB-EutMSpyC-Hag[T209C]::SpyT[588] or pCT-empty as a control were fabricated in the same manner as described above. After 24 h curing at 25 °C, the entire 1 mL solidified silica gels were dissolved in 200 μL 6x SDS-loading buffer at 100 °C for 1 h and 8 μL of spun down sample supernatant was loaded onto 15% SDS-PAGE gel.

**Flagella light microscopy**. One mL of *Bacillus* culture was centrifuged at 2,000 x *g* for 5 mins and gently resuspended in 1 mL of PBS, pH 7.5. The sample was pelleted as above, and the pellet was resuspended to an $OD_{600}$ of 10. Five μL of sample was loaded onto a polysine microscope slide, covered with a coverslip, and 10 μL Remel RYU flagella stain was applied to the edge of the coverslip.

Slides were examined using a Nikon Eclipse 90i microscope with a 100 × 1.3 numerical aperture oil-immersion objective (University Imaging Center (UIC), University of Minnesota (UMN)). Images were captured using a Nikon D2-Fi2 color camera running on Nikon's NIS-Elements software ver 5.2.1.00.

**Spore staining**. The Schaeffer and Fulton Spore stain kit was used for spore staining with a modified protocol. Briefly, 500 μL of *Bacillus* culture was placed into a glass test tube. One drop (≈150 μL) of malachite green was added to the tube and immersed into boiling water for 10 min. If samples were dehydrating before 10 min, deionized water was added dropwise.

Following boiling, 50 μL of the sample was spread by pipette onto a plain, precleaned glass slide. The slide was allowed to air dry and then heat fixed with flame. The slide was decolorized with distilled water, counterstained with safranin for ≈20 s and again decolorized.

Slides were examined using a Nikon Eclipse 90i microscope with a 100 × 1.3 numerical aperture oil-immersion objective (UIC UMN). Images were captured using a Nikon D2-Fi2 color camera running on Nikon's NIS-Elements software ver 5.2.1.00.

**SEM and TEM imaging of silica precipitation**. For SEM imaging, EutM scaffolds (1 mg/mL purified proteins in water) were attached to silicon wafers by incubating the protein suspensions on a silicon wafer for 1 h. In samples with silica, the wafers with attached protein were incubated in 100 mM silica for 2 h. Samples were washed with water, fixed with Trump's Fixative for 2 min, and washed with water again. Samples were washed with increasing ethanol concentrations of 25, 50, 75, and 100%, and then supercritically dried using a Tousimis Critical-Point Dryer (Model 780 A). Samples were sputter-coated with 8 nm of iridium and viewed with a Hitachi SU8230 field emission gun scanning electron microscope (University of Minnesota Characterization Facility).

For TEM imaging, TEM grids were coated with EutM scaffolds by incubating grids in a protein solution (0.1 mg/mL of purified His-EutM or His-EutM-CotB in water) for 10 s at room temperature. Then, a small amount of 100 mM silica was added to the grid and incubated for 1 h. Grids were then washed with deionized water, fixed with Trump's fixative for 2 mins, rewashed and stained with 2% uranyl acetate for 10 s. Grids were imaged as described below for other purified protein scaffolds.

**T209C flagella staining**. Glycerol cultures were inoculated into 4 mL of selective LB medium for plasmid maintenance and grown overnight at 30 °C and 220 rpm. One mL of culture was centrifuged at 2,000 x *g* for 10 mins. The supernatant was discarded and pelleted cells were gently resuspended in 1 mL of PBS, pH 7.5. The samples were pelleted as above and pellets resuspended with 50 μL PBS, pH 7.2 with 5 μg/mL Alexa Fluor™ 488 C$_5$ maleimide (to stain cysteines in flagella with the Hag[T209C] subunit) and incubated in the dark for five mins. 450 μL PBS was added to the samples prior to centrifugation as above. The pellet was resuspended in 50 μL PBS with 5 μg/mL SynaptoRed™ C2 (FM4-64) and incubated for five mins in the dark. Samples were pelleted as above and then resuspended with 1 mL of PBS. After another centrifugation step, the pellets were resuspended to an $OD_{600}$ of 10. Three μL of sample were mixed with an equal volume of anti-fade fluorescent mounting medium FluoroShield and applied to a polysine microscope slide.

Slides were examined using a Nikon Eclipse 90i microscope with a 100 × 1.3 numerical aperture oil-immersion objective (UIC UMN). Illumination was obtained using a Lumencor Sola Light Engine. SynaptoRed™ C2 was visualized using a dsRed fluorescence cube (excitation 530–560 nm, emission 590–650 nm) with a 3 second exposure time. Alexa Fluor™ 488 C$_5$ maleimide was visualized using a GFP fluorescence cube (excitation 450–490 nm, emission 500–550 nm) with a 5.6 second exposure time. Images were captured using a Hamamatsu Orca Flash 4.0 v2 CMOS monochrome camera in black and white using Nikon's NIS-Elements software ver 5.2.1.00.

**In situ labeling of SpyTagged flagella with His-tdTomato-SpyCatcher**. Glycerol cultures were inoculated into 4 mL of selective LB medium containing 0.15 mg of purified His-tdTomato-SpyCatcher protein and grown overnight at 30 °C and 220 rpm. One mL of culture was centrifuged at 2000 x $g$ for 10 mins and gently resuspended in one mL of PBS, pH 7.5. The samples were pelleted as above. The pellets were resuspended in 50 µL PBS pH 7.2 with 5 µg/mL Alexa Fluor$^{TM}$ 488 C$_5$ maleimide and incubated in the dark for five mins. 450 µL PBS was added to the samples prior to centrifugation as above. Cellular nucleic acid was stained by resuspending the pellets in 50 µL Tris-HCl, pH 8.0 with 10 µM Syto16 and incubated in the dark for 10 mins. 450 µL Tris-HCl, pH 8.0 was added and samples were centrifuged as above. Unused dye was removed by resuspension with 1 mL PBS and samples were centrifuged again. After resuspension to an OD$_{600}$ of 10, 3 µL of sample was mixed with an equal volume of FluoroShield and loaded onto a polysine microscope slide.

Slides were examined using a Nikon Eclipse 90i microscope with a $100 \times 1.3$ numerical aperture oil-immersion objective (UIC UMN). Illumination was obtained using a Lumencor Sola Light Engine. tdTomato was visualized using a dsRed fluorescence cube (excitation 530–560 nm, emission 590–650 nm) with a 3 second exposure time. Alexa Fluor$^{TM}$ 488 C$_5$ maleimide and Syto$^{TM}$16 were visualized using a GFP fluorescence cube (excitation 450–490 nm, emission 500–550 nm) with a 0.3–5.6 s exposure time. Images were captured using a Hamamatsu Orca Flash 4.0 v2 CMOS monochrome camera in black and white using Nikon's NIS-Elements software ver 5.2.1.00.

**TEM of purified scaffolds from *E. coli* and *Bacillus***. Purified His-EutM-SpyCatcher protein scaffolds were diluted to 1 mg/mL. Ten µL was placed onto 200 mesh copper grids with 10 nm formvar and 1 nm carbon for 3 mins. The fluid was removed and 10 µL of Trump's fixative (4% formalin, 1% glutaraldehyde, 0.1 M sodium cacodylate) was applied for 5 mins to fix the proteins. The fluid was removed and the grids were rinsed with 10 µL of deionized water three times. Negative staining was completed using 1% aqueous uranyl acetate which was applied and removed immediately to prevent over-staining. Grids were examined using a JEOL-JEM-1400Plus transmission electron microscope with a LaB6 tungsten filament at 60 kV (UIC UMN). Images were acquired using an Advanced Microscopy Techniques XR16 camera using AMT capture Engine software version 7.0.0.187.

**In situ labeling of secreted EutM-SpyCatcher scaffolds with His-SpyTag-eGFP**. *B. subtilis* Δ*lyt*C Δ*flh*G harboring pCT-EutMCotB-EutMSpyC-Hag$^{T209C}$::SpyT$^{588}$ was grown under standard conditions with (induced) and without (uninduced) cumate added for induction of protein expression. 500 µL of culture was mixed with 0.25 mg His-eGFP or His-SpyTag-eGFP (an estimated 2:1 ratio of SpyCatcher: eGFP established in[35]) and mixed at RT for 30 mins. Ten µL was applied to a glass slide and covered with a slip. Slides were examined using a Nikon Eclipse 90i microscope with a $40 \times 0.75$ numerical aperture objective (UIC UMN). Illumination was obtained using a Lumencor Sola Light Engine. His-eGFP and His-SpyTag-eGFP were visualized using a GFP fluorescence cube (excitation 450–490 nm, emission 500–550 nm) with a 0.032–0.2 second exposure time. DIC images were captured with a 0.037–0.039 second exposure. Images were captured using a Hamamatsu Orca Flash 4.0 v2 CMOS monochrome camera in black and white using Nikon's NIS-Elements software ver 5.2.1.00.

**Silica material thin sections and TEM**. *B. subtilis* Δ*lyt*C Δ*flh*G harboring pCT-EutMCotB-EutMSpyC-Hag$^{T209C}$::SpyT$^{588}$ was grown under standard conditions with (induced) and without (uninduced) cumate added for induction of protein expression. Cultures with cell densities of OD$_{600}$ ~3.5 (induced) or ~6 (uninduced) were either directly silica biomineralized or were concentrated prior to silica gel formation. For concentration, cultures were centrifuged at 3,220 x $g$ for 10 mins and the supernatant was removed. Pellets were resuspended to OD$_{600}$ 10 or 20 using the media removed from the pellet. Silica gel plugs were prepared as described above for mechanical testing using 200 mM silica. Gels were pushed out of the syringes and 2 mm$^3$ gel portions were cut from the center of both induced and uninduced samples. The gel pieces were fixed at 4 °C and 10 rpm overnight in 100 mM sodium cacodylate, 2% glutaraldehyde, and 2% paraformaldehyde. The gel pieces were then rinsed with 100 mM sodium cacodylate three times for 30 mins each at 4 °C and 10 rpm. An overnight post-fixation using 4% osmium tetraoxide in 100 mM sodium cacodylate was completed at 4 °C and 10 rpm. The samples were extensively rinsed with water and subjected to an ethanol dehydration series (25, 50, 75, 95, and 100%). Gel samples were trimmed into approximately 1 mm$^3$ pieces, infiltrated with EMbed 812 resin (1:2 resin:ethanol, 1:1, 2:1, 100% resin without hardener, 100% resin with benzyl dimethyl amine hardener repeated once (2x total); all times 8 h-overnight), and polymerized in a 60 °C oven for 48 h. Sections were cut using a diamond knife on a Leica Ultracut UCT microtome at a thickness of 70–100 microns and collected on 200-mesh formvar/carbon-coated copper grids. Sections were stained with 3% aqueous uranyl acetate for 20 mins, rinsed in ultrapure water (5 sec, 5x), stained with Sato's triple-lead stain[80] for 3 mins, and rinsed in ultrapure water (5 sec, 5x). Samples were examined using a JEOL-JEM-1400Plus transmission electron microscope with a LaB6 tungsten filament at 60 kV (UIC UMN). Images were acquired using an Advanced Microscopy Techniques XR16 camera using AMT capture Engine software version 7.0.0.187.

**Image Analysis**. Images were cropped, scale bars added, and channels were colorized and merged using Fiji v. 1.53[81]. Supplementary Fig. 9 image edits for part A include *B. subtilis* Δ*lytC* Δ*flhG* strain SpyTag 555 in which the red and green channels were multiplied by 1.25 before merging. Supplementary Fig. 12 image edits for part A include the Hag$^{T209C}$ control in which the red channel was multiplied by 1.5 before merging with the green channel. In addition, for Hag$^{T209C}$::SpyTag$^{588}$ the red channel was multiplied by 1.25 before merging. Supplementary Fig. 12 image edits in part B include the Hag$^{T209C}$ control in which the red channel was multiplied by 1.5 before merging. In addition, for Hag$^{T209C}$::SpyTag$^{588}$ the red channel was multiplied by 1.25 before merging.

Flagella RYU staining images were converted to grayscale and had the contrast and brightness increased by 10 using GNU Image Manipulation Program (GIMP) v. 2.8.18.

**Statistical analysis**. Statistical analysis of OD and pH data was performed using Origin v. 9.1. Statistical analysis of rheology data was performed using Microsoft Excel 2016. Mean and standard deviations from three biological replicates were calculated using Average and STDEV functions. Standard deviations from three biological replicates are shown as error bars in graphs and plots.

Statistical significance ($p$-values) of rheology data (Supplementary Table 2) was determined by performing T-test (Paired Two Sample for Means) and ANOVA (One-way, Single-factor) analysis with G' (storage modulus) and G'' (loss modulus) angular frequency averages. T-Test analysis was used to determine the statistical significance of rheology data between two groups of angular frequency data (G' or G'') from silica gels fabricated with *B. subtilis* strains transformed with three different plasmids (pCT-empty, pCT-EutMCotB-EutMSpyC, pCT-EutMCotB-EutMSpyC-HagT209C::SpyT588). ANOVA analysis was performed on three groups of angular frequency data (G' or G'') from silica blocks fabricated by the same three *B. subtilis* strains. Data were considered statistically significant when $p < 0.05$.

**Reporting Summary**. Further information on research design is available in the Nature Research Reporting Summary linked to this article.

## Data availability

All data generated or analyzed in this study are included in this article and accompanying Supplementary Information, Supplementary Data and Source Data files. Engineered *Bacillus* strains and plasmids created in this study can be made available subject to an MTA that can be requested by contacting the corresponding author Prof. Schmidt-Dannert (schmi232@umn.edu), who will respond to requests within a week. Protein Data Bank data sets (PDB: 5WJX, PDB: 6GOW, PDB: 4MLI) were used to generate protein structure representations. Source data are provided with this paper.

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

## Acknowledgements

This research was sponsored by the Defense Advanced Research Projects Agency (DARPA contract HR0011-17-2-0038) and supported by seed grants from the Biotechnology Institute at the University of Minnesota. Parts of this work were carried out in the University of Minnesota's Characterization Facility, which receives partial support from the NSF through the MRSEC (Award Number DMR-2011401) and NNCI (Award Number ECCS-2025124) programs. We also acknowledge resources and staff at the University of Minnesota core facilities for contribution to this work: Imaging Center (Dr. Gail Celico, sample thin sectioning) and Center for Mass Spectrometry (LC-MS sample analysis, Dr. LeeAnn Higgins and Todd Markowski).

## Author contributions

S.-Y.K. created and tested expression constructs for scaffold secretion and silica biomineralization in *B. subtilis*, designed and performed silica biomineralization experiments and analyzed data, wrote manuscript and Supplementary information. A.P. engineered constructs for endospore formation and SpyTag display into *B. subtilis* strain, optimized and analyzed scaffold secretion, wrote manuscript and Supplementary information. S.B. developed and optimized methods for strain phenotype analysis by light/fluorescence microscopy, performed the microscopic analysis of engineered *B. subtilis* cultures to visualize flagella and endospore. S.B. performed Bacillus silica material analysis by electron microscopy, wrote manuscript and Supplementary information. J.J.B. characterized silica condensation by protein scaffolds using spectrophotometric assays and by SEM and TEM and wrote methods for this characterization work. J.J.B. proof-read and commented on manuscript. S.O.S. selected locations for the flagella SpyTag display and designed initial set of constructs for *B. subtilis* engineering. S.S. proof-read and commented on manuscript. M.B.Q. designed and purified scaffolds for silica biomineralization and contributed to design, planning, and supervision of the work, M.B.Q. proof-read and commented on manuscript. A.A. contributed to the design, planning, and supervision of the silica biomineralization experiments. A.A. provided feedback and comments on the manuscript. C.S.-D. conceived, devised, conceptualized, and directed the overall project, analyzed data and wrote the manuscript draft together with S.-Y.K., A.P., S.B.

## Competing interests

The authors declare no competing interests.
