## [Peer Review File · Nature Communications]

Engineering *Bacillus subtilis* for the formation of a durable living biocomposite materialReviewers' Comments:

Reviewer #1:

Remarks to the Author:

The work described in this article represents a proof of concept for the fabrication of ELMs based using *Bacillus subtilis* and EutM proteins as self-assembling building blocks. It is an interesting study, and the choice of strain (*B. subtilis*) and building blocks (EutM) is valid. However, I found the justification of these choices to be lacking, and the rationale for choosing silica as mineral component of the materials to be missing.

The EutM building block was justified by the authors as being able to tolerate N- and C-terminal mutations, and thus being able to form a fusion with SpyCatcher. But how does it compare with other self-assembling protein systems / protein fibers that have been used previously in ELMs? Why use the EutM system over other proteins (E.g. CsgA, TasA, others?)? Why is the hexameric structure of particular interest? And why using SpyTag-SpyCatcher assembly for silica-binding peptides rather than simply fusing the silica peptide to EutM proteins?

Bacillus subtilis has been shown to survive under extreme conditions and to be able to enter a dormant spore state or a germination state depending on the environmental conditions, which is certainly of interest for ELMs. However, this was not mentioned clearly by the authors, and the choice of the strain is not justified. Since *B. subtilis* was used, it would have been interesting to capitalize further on spore-forming ability.

The authors simply state (lines 58-64) that they wish to "broaden the ELM landscape". This does not appear like a strong rationale for undertaking the work. It is also not clear why silica was chosen as the mineral component of this ELM. What are potential applications of this composite ELM material? Apart from writing a better justification for their author, I found that the authors presented the methods and results clearly. I did not notice flaws in result interpretation and conclusions. Novel results of interest include the test for signal sequences for secreting building blocks from *Bacillus subtilis*, which could be applied to the secretion of other proteins from ELMs. The change in mechanical properties of the material over time and the regeneration of the biocomposite are also concepts of interest for the ELM community.

Clarifications could however be presented for the silica precipitation assay using molybdate blue, to explain how the assay is performed. Silica precipitation was also only observed via SEM. Could TEM analysis be used to provide better structural information on the minerals formed? Would authors expect to be able to control silica morphology at the nanoscale rather than the microscale using the EutM system?

Reviewer #2:

Remarks to the Author:

The manuscript by Kang, Pokhrel, Bratsch et al. is an exciting demonstration of an Engineered Living Material (ELM) produced from growing engineered *Bacillus* cells and silica. Cells are engineered to secrete modular proteins based on the microcompartment shell protein, EutM, that can be made to fuse to the bacillus flagella and to biomineralization peptides. The result is a culture of bacillus that grows and interconnects covalently between cells and flagella to become a living scaffold rich in biomineralization sites. When silica is added to this (via hydrolyzed TEOS) a gel-like material quickly forms due to biomineralization. This is big enough after an hour to be a visible material that can be held and manipulated. Impressively the authors also engineer the bacillus to produce endospores within the cells, so that when the material is damaged and cells rupture, spores can be germinated for material regeneration. As a proof of concept they also engineer the cells to express purple chromoproteins and this leads to a purple material. Very cool!

I was very impressed by this manuscript. It amounts to a huge body of work and as such takes quite a while to read through and is rich with tons of data. Often that can make a paper hard to read and hard to follow, but this one is very well written and presented and the authors should be congratulated on

such good work. From a synthetic biology and ELMs perspective (my expertise) I see very little issues with the paper and its hard to even find minor corrections. It is true that in many places the engineering efforts don't give huge improvements to the final material properties (Fig 6C for example), but the small improvements all add up a bit, and the actually toughness of the material produced is not necessarily a big deal because the material can also do some level of regeneration and can be engineered to do lots of other cool things too - like being purple. I shared the manuscript with a member of my team who is an expert on protein secretion and they were also impressed, but mentioned that some of the secretion assays (blots?) needed better controls. I didn't get any more specifics as she has gone on leave, but I thought I would mention this in case it is a concern of another review. If not, then forget it.

In the end I only have 1 request for improving/clarifying the work...

1. On Figure 4, panel G is supposed to show a band of the EutM complex in lane 2 (it is indicated by an arrow). In my print-out of the paper it is very hard to see anything here. Can this image be improved or replicated?

Reviewer #3:

Remarks to the Author:

Review of 'Engineering Bacillus subtilis for the formation of a durable living biocomposite material' Kang et al.

This manuscript is a comprehensive study that combines molecular biology and materials chemistry. It showcases the development of a particular ELM 'chassis' towards the development of 'living' biocomposites though it is a great pity that the 'scaffold building block' component was not viable two weeks after encasing in a the silica polymer matrix. It is not clear to me WHO the authors intend as their audience. For the widest visibility of their work it needs to include scientists and engineers who do not have molecular biology as their background. I do think it is possible to make the research more 'friendly' to a wider audience than at present – perhaps by explaining some of the terminology and being more careful with the terms and schematics used. At present the MS (and SI) is a story where each step along the way is described (in considerable detail). Although the material is necessary to establish the success (or otherwise) of each stage the MS could do with a rethink as to the MINIMUM required to tell and validate the MAIN research findings. I believe this would enhance its reach and impact.

In response to the required questions at review. Note, my detailed comments, questions and matters for clarification will be listed in detail below these general responses.

1. The research contains noteworthy results- specifically in the molecular biology in establishing the specific mutations required to engineer a nominally 'living material'. Further, growth in the presence of a silica condensation system, although not of itself novel is a useful element of the study.
2. The research is of potential significance to the molecular biology and materials engineering communities- though for the latter, 'simple' is better and the current system- if designed to be 'directional'- does not show much evidence for this.
3. Results are generally supportive of the conclusions/ hypotheses proposed though there are some areas where additional evidence is needed. Figure legends are to long and contain results and discussion and need simplifying.
4. Most of the data analysis is good but there are elements of the microscopy studies and materials characterisation that require statistical analysis (microscopy and mechanical analysis), quantitation (microscopy of organismal growth- flagella type etc), higher magnification images to support the statements made (particularly boundaries around cells in sections), EDX to identify silica.
5. Methodology, carefully described in the SI is generally sound though care must be taken to fully describe chemical procedures.
6. See above, in general, sufficient detail is provided.

Detailed comments:

1. Abstract- regeneration was NOT from a piece of silica material- it was from a piece of silica material containing cells- please add a couple of words to this effect. Mechanical properties- see later- please report values compared to the control and include statistics- are they statistically different, either to the control or between the different constructs? Clarify. What do you mean by 'responsive coatings (functional coatings) and plasters'? These applications appear in several places within the MS but I don't believe you say anything about the links. Even if these are tentative- what is it about your materials that might make them useful for these applications?
2. Figure 1 really looks like a TOC- it is not sufficiently referred to in the TEXT. If you want to keep it then refer to it more broadly as you discuss the plan of the paper and ensure it matches. At present it is difficult to follow without reading the MS first.
3. General introduction sets the scene for the research well. Line 82- what do you mean by 'cluster, polar flagellar'? I know what you mean from reading the whole paper but perhaps an explanation with a few extra words would help? There is definitely at least one word missing here.
4. Line 97- why does it matter if a discolored material results?
5. Line 107- alkalifies is a made up word (at least to this reviewer) -please choose a technically more correct word.
6. Line 124 put definition for peritrichous in brackets to help the reader – (or earlier if you use this word beforehand).
7. Figure 2- the same is true for all legends- far too wordy- for example 'Samples analysed included', title of 2b is a result NOT a legend. Some figure legends are much worse- all need to be fixed.
8. Figure 3. Part A- very confusing- the effect of the second deletion is clear- the effect of the first deletion- something is 'added' to the figure- what does this mean? I understand what you are doing by making the deletion, but the figure is counterintuitive. Likewise, the positioning of the words and brackets above and below the 3 images is a little confusing. Please reorganise. Data for flagella phenotyping are not convincing (at least to this referee)- I would expect to see more quantitative data rather than a couple of images which even at the max magnification on a good computer were far from convincing).
9. Line 196-207. Is this relevant for the main story? Wouldn't this be better in the SI and perhaps a sentence only in the main text- if at all.
10. Evidence for rolled up tubes-not evident from the data presented- again- what in the paragraph starting line 210 is relevant for the main story?
11. Figure 4b- nice if you like images of proteins and their active site- but- is it only relevant here if you discuss the behaviour of the different mutants in relation to their position/ accessibility? This is not done but could be.
12. Figure 4d- is this necessary in the main paper? You don't really make much reference to it in the text. Legends for this figure- far too long.
13. Line 245 onwards. Use of the word 'some' 'best' 'shorter' all qualitative and not entirely convincing. If you have the data make it more robust. If you don't need really need anything other than a qualitative 'check' then say such and move on.
14. Line 255. Explain why it is important that your construct can make 'isopeptide' bonds (think of your intended audience).
15. Line 268. Ensure that Fig. S10a and related figures includes the His tag domain to be clear HOW the various components are connected. You allude to the His tag in the main text but the supplementary figure is missing information.
16. Good to see that you use well controlled silicifying solutions though note- at the concentrations you are using condensation will start as soon as the precursor starts to hydrolyse- so, to be consistent with all of the excellent experimental descriptions- please include in your experimental the time (80 mins?) after the start of hydrolysis that you added your silicifying solutions to your cell media. This can be added to the SI.
17. Fig. 5a, S10b- 'rolled up tubes' scale of images is not helpful to see this well- Energy dispersive x-ray analysis is needed to confirm the presence of silica. From the scale of the images-it is not easy to see the increase in roughness you state in the text. You could describe the materials in terms of particle sizes- does silica addition affect this- HOW much? Does silica addition generate larger

structures overall suggesting a coating or do the structures shrink? Even with the data you have you could say more that would be useful in terms of taking the materials forward. What properties of your composites will be useful?

18. Line 305/ and or Figure S11 etc- add culture concentrations (however you want to refer to this- there is no information in the SI for these expts). Associated figure S12 etc- I did not find convincing. Is the section from Line 317 relevant to the main story?

19. Figure 6. S13 and associated Text. 'Larger' etc- measure and provide semi-quantitative data.

20. Rheological measurements- as a minimum the numbers require X plus/minus SD but you can test whether they are statistically significantly different and report the same in the abstract. Data can be used from Table S6.

21. The data to support the location of the secreted scaffolds 7a is convincing, that of 7b and images in the SI less so. I would have expected to see images at much higher magnification than presented here.

22. Line 380 onwards. A 24 hour sample is NOT dry and the silica component of the material will still be condensing. Again- you refer to 'functional coatings or plasters' 'integrate desired functions'- what are these? Not clear how you do these experiments- do you break down the silica material- if not- do you remove it after a certain time? - how much 'Si' gets into solution etc before adding 'more' hydrolysed TEOS. A great pity that certain of the mutations do not survive to 2 weeks- though good to see you have a strategy to overcome this.

23. Line 425. Be careful in your use of the word silica' - you add a mix of silica oligomers/ polymers (depending on concentration used as a minimum) so please rethink whether this word is appropriate.

24. In general, figures and legends need modifying- see examples above. Some further data analysis is needed to properly support some of the statements made.

Response to Reviewers Comments:

We thank the reviewers for their constructive reviews and are providing our point-by-point responses below.

Reviewer #1:

The work described in this article represents a proof of concept for the fabrication of ELMs based using *Bacillus subtilis* and EutM proteins as self-assembling building blocks. It is an interesting study, and the choice of strain (*B. subtilis*) and building blocks (EutM) is valid. However, I found the justification of these choices to be lacking, and the rationale for choosing silica as mineral component of the materials to be missing.

Response: We have added justifications for the use of the EutM building blocks and silica for biomineralization in the introduction.

The EutM building block was justified by the authors as being able to tolerate N- and C-terminal mutations, and thus being able to form a fusion with SpyCatcher. But how does it compare with other self-assembling protein systems / protein fibers that have been used previously in ELMs? Why use the EutM system over other proteins (E.g. CsgA, TasA, others?)?

Response: We have added the following two sentences to the introduction to explain our reason for using a different self-assembling system rather than the bacterial amyloid fibers typically engineered for biofilm formation: *“We rationalized that 2D-scaffold forming proteins rather than the commonly engineered bacterial biofilm-associated amyloid fibers (e.g. CsgA from *E. coli* or TasA from *B. subtilis*¹⁰) will form different matrix architectures and surfaces for functionalization. Demonstration of extracellular matrix formation from non-amyloid protein building blocks will also lay the foundation for the design of new types of ELM matrices from the many other protein building blocks currently assembled into functional bionanomaterials³⁶.”*

Why is the hexameric structure of particular interest?

Response: It is not the hexameric structure per se that is of particular interest. We stated: *“Our 2D-scaffolds self-assemble from hexameric units of the bacterial microcompartment shell protein EutM”*. As explained above, 2D-scaffolds formation from non-amyloid building blocks is of interest.

And why using SpyTag-SpyCatcher assembly for silica-binding peptides rather than simply fusing the silica peptide to EutM proteins?

Response: We are not sure what exactly the reviewer means here. The SpyTag-SpyCatcher system is used for cell attachment. We created two types of EutM building blocks: One EutM building block containing a C-terminal SpyCatcher fusion for cell attachment, and another EutM building block containing a C-terminal CotB biomineralization peptide fusion. Both EutM building blocks already contained an N-terminal His-tag and a secretion signal sequence fusion. Although it would have been possible to also add the CotB peptide to the N-terminus of the His-EutM-SpyCatcher building block, we reasoned that creating two different building blocks is simpler and offers for future work more flexibility for separately tuning the expression of different building blocks.

Bacillus subtilis has been shown to survive under extreme conditions and to be able to enter a dormant spore state or a germination state depending on the environmental conditions, which is certainly of interest for ELMs. However, this was not mentioned clearly by the authors, and the choice of the strain is not justified. Since *B. subtilis* was used, it would have been interesting to capitalize further on spore-forming ability.

Response: We have added the following sentences to the introduction:

“Importantly, it forms spores that remain viable for a long time and allow the bacteria to survive extreme conditions²⁹⁻³¹. B. subtilis cells will therefore be able to enter a dormant spore state in our ELM under unfavorable environmental conditions, allowing it to persist until favorable conditions induce germination and cell revival.”

“In addition, because spores contain the genetic information that was programmed into engineered vegetative cells, living materials may be autonomously fabricated at the sites of use from stored spores.”

The authors simply state (lines 58-64) that they wish to “broaden the ELM landscape”. This does not appear like a strong rationale for undertaking the work. It is also not clear why silica was chosen as the mineral component of this ELM. What are potential applications of this composite ELM material?

Response: See our response above for justifying the use of our protein scaffolding system. We also mentioned that the formed scaffolds are highly robust, which is important for creating a durable material. We have added two sentences to explain why we have chosen silica for biomineralization.

“As a proof-of concept, we chose silica, as one of the most abundant earth minerals that is inexpensive and the main ingredient of many building materials. A number of proteins are also known to control the biomineralization of orthosilicic acid $\text{Si}(\text{OH})_4$ (the soluble form of silica), which can be used as a source for biomineralization peptides³⁷.”

Apart from writing a better justification for their author, I found that the authors presented the methods and results clearly. I did not notice flaws in result interpretation and conclusions. Novel results of interest include the test for signal sequences for secreting building blocks from Bacillus subtilis, which could be applied to the secretion of other proteins from ELMs. The change in mechanical properties of the material over time and the regeneration of the biocomposite are also concepts of interest for the ELM community.

Response: We thank the reviewer for their overall positive comments on the significance and clarity of the presented results.

Clarifications could however be presented for the silica precipitation assay using molybdate blue, to explain how the assay is performed.

Response: The silica precipitation assay was described in much detail in the methods section of the supporting information. Unless we misunderstood this comment, we think the experimental details are sufficient: *“Silica precipitation was measured using the blue silicomolybdate method^{17,18}. Silica was added to purified protein scaffolds, at a concentration of 1 mg/mL in deionized water, to a concentration of 100 mM and the suspension was mixed for 2 h at room temperature (RT). The precipitate was then collected by centrifugation (3 mins at 18,800 x g), and the resulting pellet was washed 3 times with deionized water. Precipitated silica in the scaffold pellet was depolymerized by adding 1 M NaOH for 10 mins. The samples were then diluted to 1.6 mL and 0.15 mL of solution A (containing 20 g/L ammonium molybdate tetrahydrate and 60 mL/L concentrated hydrochloric acid in deionized water) was added to the sample. After 10 mins, 0.75 mL of solution B (containing 20 g/L oxalic acid, 6.67 g/L 4-methylaminophenol sulphate, 4 g/L anhydrous sodium sulfite, and 100 mL/L concentrated sulphuric acid in deionized water) was added. The blue color was allowed to develop for 2 h at RT, and the absorbance was measured at 810 nm. All samples were run in triplicate.”*

We added a statement to the section on the “Preparation of hydrolyzed TEOS as silica source” to make clear that this was used a silica source in silica binding and biomineralization experiments (see also response to reviewer 3).

Silica precipitation was also only observed via SEM. Could TEM analysis be used to provide better structural information on the minerals formed?

Response: Prior to SEM analysis of silica precipitation on scaffold, we did characterize silica precipitation on scaffolds by TEM. SEM imaging, however, provided much clearer visualization of silica nano-particle formation on the scaffolds. But as suggested by the reviewer, we now describe TEM imaging results for His-EutM and His-EutM-CotB scaffolds in the manuscript (p.11 in the revision) and show TEM images in Supporting Fig. 10c.

Would authors expect to be able to control silica morphology at the nanoscale rather than the microscale using the EutM system?

Response: Controlling biomineralization (and other functions) at the nanoscale on our scaffolds is an area of research that we are currently exploring by assembling in vitro different, functionalized scaffold building blocks. We believe that by spatially controlling surface functionalization on EutM hexameric array it will eventually become possible to control in a predictable manner biomineralization. This is ongoing work aimed at designing functional protein biocomposite materials that is outside the scope of this work which is on the formation of a living material.

Reviewer #2 (Remarks to the Author):

The manuscript by Kang, Pokhrel, Bratsch et al. is an exciting demonstration of an Engineered Living Material (ELM) produced from growing engineered Bacillus cells and silica. Cells are engineered to secrete modular proteins based on the microcompartment shell protein, EutM, that can be made to fuse to the bacillus flagella and to biomineralization peptides. The result is a culture of bacillus that grows and interconnects covalently between cells and flagella to become a living scaffold rich in biomineralization sites. When silica is added to this (via hydrolyzed TEOS) a gel-like material quickly forms due to biomineralization. This is big enough after an hour to be a visible material that can be held and manipulated. Impressively the authors also engineer the bacillus to produce endospores within the cells, so that when the material is damaged and cells rupture, spores can be germinated for material regeneration. As a proof of concept they also engineer the cells to express purple chromoproteins and this leads to a purple material. Very cool!

I was very impressed by this manuscript. It amounts to a huge body of work and as such takes quite a while to read through and is rich with tons of data. Often that can make a paper hard to read and hard to follow, but this one is very well written and presented and the authors should be congratulated on such good work. From a synthetic biology and ELMs perspective (my expertise) I see very little issues with the paper and its hard to even find minor corrections. It is true that in many places the engineering efforts don't give huge improvements to the final material properties (Fig 6C for example), but the small improvements all add up a bit, and the actually toughness of the material produced is not necessarily a big deal because the material can also do some level of regeneration and can be engineered to do lots of other cool things too - like being purple. I shared the manuscript with a member of my team who is an expert on protein secretion and they were also impressed, but mentioned that some of the secretion assays (blots?) needed better controls. I didn't get any more specifics as she has gone on leave, but I thought I would mention this in case it is a concern of another review. If not, then forget it.

Response: We thank this reviewer for their enthusiastic and very positive review of our manuscript. We are not sure what their team member means with "better controls" for secretion assays. All SDS-PAGE gels included appropriate control samples and we also confirmed secreted proteins by LC-MS analysis (Supporting Fig. 7).

In the end I only have 1 request for improving/clarifying the work...

1. On Figure 4, panel G is supposed to show a band of the EutM complex in lane 2 (it is indicated by an arrow). In my print-out of the paper it is very hard to see anything here. Can this image be improved or replicated?

Response: We agree this band was indeed difficult to see in Figure 4 when printed. We have increased the color intensity of the image so that this band is now better visible (all gel images are provided as raw images as well). The reason why this band is relatively weak compared to the major flagellin band is that we co-express the modified flagellin hag protein in the context of the native flagellin hag proteins expressed in *B. subtilis*. The flagella therefore contain only a small number of SpyTag-displaying flagellin monomers. We have added a sentence to explain this in the manuscript (p. 10).

Reviewer #3

Review of 'Engineering Bacillus subtilis for the formation of a durable living biocomposite material'

Kang et al.

This manuscript is a comprehensive study that combines molecular biology and materials chemistry. It showcases the development of a particular ELM 'chassis' towards the development of 'living' biocomposites though it is a great pity that the 'scaffold building block' component was not viable two weeks after encasing in a the silica polymer matrix.

It is not clear to me WHO the authors intend as their audience. For the widest visibility of their work it needs to include scientists and engineers who do not have molecular biology as their background. I do think it is possible to make the research more 'friendly' to a wider audience than at present – perhaps by explaining some of the terminology and being more careful with the terms and schematics used.

At present the MS (and SI) is a story where each step along the way is described (in considerable detail). Although the material is necessary to establish the success (or otherwise) of each stage the MS could do with a rethink as to the MINIMUM required to tell and validate the MAIN research findings. I believe this would enhance its reach and impact.

Response: When we initially wrote drafts of this manuscript, we had to decide of either writing a very succinct manuscript or a more detailed step-by-step manuscript. In the end, we decided to go with the more detailed manuscript because of the amount of data that needed to be presented in supporting our ELM design, which required substantial engineering, optimization and characterization efforts. Reviewers 1 and 2 agree with our decision: Reviewer 1 comments that the “authors presented the methods and results clearly. I did not notice flaws in result interpretation and conclusions.” Reviewer 2 states: “I was very impressed by this manuscript. It amounts to a huge body of work and as such takes quite a while to read through and is rich with tons of data. Often that can make a paper hard to read and hard to follow, but this one is very well written and presented and the authors should be congratulated on such good work.”

We feel that the manuscript will appeal both to the synthetic biology community and to material scientists interested in bioengineering. Leaving out details and the rational used for strain engineering and testing of desired phenotypical outcomes, including protein secretion, would have made it much less interesting for the synthetic biology and ELM community. Co-author Prof. Aksan is a mechanical engineer, material scientist and an expert on silica materials, who strongly favored a step-by-step “story” for clarity reasons – in particular, for readers not so familiar with genetic and microbial engineering.

We have explained some molecular biology details on e.g. the use of the SpyTag-Catcher system, more clearly.

In response to the required questions at review. Note, my detailed comments, questions and matters for clarification will be listed in detail below these general responses.

1. The research contains noteworthy results- specifically in the molecular biology in establishing the specific mutations required to engineer a nominally 'living material'. Further, growth in the presence of a silica condensation system, although not of itself novel is a useful element of the study.

2. The research is of potential significance to the molecular biology and materials engineering communities- though for the latter, 'simple' is better and the current system- if designed to be 'directional'- does not show much evidence for this.

Response: See comments above. We do not follow what this reviewer means with 'directional'. It is "directional" in a sense that we accomplished the functions that we set out to obtain through our initial design strategy shown in Fig. 1.

3. Results are generally supportive of the conclusions/ hypotheses proposed though there are some areas where additional evidence is needed. Figure legends are too long and contain results and discussion and need simplifying.

Response: We have shortened the legends when possible. The legend of Fig. 4 has been significantly shortened. It is our belief that figure legends should largely be self-explanatory.

4. Most of the data analysis is good but there are elements of the microscopy studies and materials characterisation that require statistical analysis (microscopy and mechanical analysis), quantitation (microscopy of organismal growth- flagella type etc), higher magnification images to support the statements made (particularly boundaries around cells in sections), EDX to identify silica.

Response: See our responses below. We have added a higher magnification image to show cell boundaries in the silica material, added TEM images (see also our response to reviewer 1) and included statistical analysis of the rheology data.

5. Methodology, carefully described in the SI is generally sound though care must be taken to fully describe chemical procedures.

Response: We are not sure which chemical procedures are lacking descriptions. We are assuming that the reviewer is referring to the method description for Supporting Fig. 11. We have now added a detailed description under "Optimizing silica concentrations" in the Supporting Information.

6. See above, in general, sufficient detail is provided.

Detailed comments:

1. Abstract- regeneration was NOT from a piece of silica material- it was from a piece of silica material containing cells- please add a couple of words to this effect. Mechanical properties- see later- please report values compared to the control and include statistics- are they statistically different, either to the control or between the different constructs? Clarify. What do you mean by 'responsive coatings (functional coatings) and plasters'? These applications appear in several places within the MS but I don't believe you say anything about the links. Even if these are tentative- what is it about your materials that might make them useful for these applications?

Response: As suggested, we have changed a "piece of silica material" to "piece of silica material containing cells". We have also modified the last sentence in the introduction to better explain "responsive coatings and plasters". It now reads: "self-healing materials for use as coatings and plasters that can respond to external stimuli due to the functions provided by the engineered cells in such materials." Note: there is a word limit of 150, so we could not include more information in the abstract – in fact, we had to shorten it. We included this information therefore in the introduction.

2. Figure 1 really looks like a TOC- it is not sufficiently referred to in the TEXT. If you want to keep it then refer to it more broadly as you discuss the plan of the paper and ensure it matches. At present it is difficult to follow without reading the MS first.

Response: We have expanded the legend for Fig. 1 and refer to Fig. 1 throughout the manuscript.

3. General introduction sets the scene for the research well. Line 82- what do you mean by 'cluster, polar flagellar'? I know what you mean from reading the whole paper but perhaps an

explanation with a few extra words would help? There is definitely at least one word missing here.

Response: A word was indeed missing, it now reads “*SpyTags on clusters of polar flagella*”.

4. Line 97- why does it matter if a discolored material results?

Response: This refers back to its use as coating and plasters. We modified this sentence to: “*do not create an overly discolored material for aesthetic and functional (for example for sentinel or stealth use) reasons*”

5. Line 107- alkalifies is a made up word (at least to this reviewer) -please choose a technically more correct word.

Response: Changed to “turns increasingly alkaline”

6. Line 124 put definition for peritrichous in brackets to help the reader – (or earlier if you use this word beforehand).

Response: As suggested, we have added “(uniformly distributed flagella)” after peritrichous flagella.

7. Figure 2- the same is true for all legends- far too wordy- for example ‘Samples analysed included’, title of 2b is a result NOT a legend. Some figure legends are much worse- all need to be fixed.

Response: As stated earlier, we have shortened the figure legends as much as possible while still making sure that the figures remain largely self-explanatory.

8. Figure 3. Part A- very confusing- the effect of the second deletion is clear- the effect of the first deletion- something is ‘added’ to the figure- what does this mean? I understand what you are doing by making the deletion, but the figure is counterintuitive. Likewise, the positioning of the words and brackets above and below the 3 images is a little confusing. Please reorganise. Data for flagella phenotyping are not convincing (at least to this referee)- I would expect to see more quantitative data rather than a couple of images which even at the max magnification on a good computer were far from convincing).

Response: We have modified Fig. 3a as suggested to make the phenotypes clearer.

We believe that Fig. 3b clearly shows spore release by WT *B. subtilis* and containment of endospores in the engineered strain. Likewise, peritrichous flagellation is clearly visible for the WT strain compared to the formation of polar, clustered flagella by the engineered strain. We do not believe there is a need for quantifying flagella phenotypes. Flagellation of WT *B. subtilis* is well documented in the literature, as is the phenotype of the $\Delta flhG$ mutation (referenced in this manuscript). Additional microscopy images are provided in the Supporting Figures for flagella phenotypes of the WT and engineered strains.

9. Line 196-207. Is this relevant for the main story? Wouldn’t this be better in the SI and perhaps a sentence only in the main text- if at all.

Response: These results are critical for establishing that the scaffold building blocks are secreted. It also provides important information for the secretion of other self-assembling proteins by *B. subtilis* for the fabrication of different types of ELMS (see also our response to reviewer 1, and remarks by reviewer 2 that comment on the importance of protein secretion).

10. Evidence for rolled up tubes-not evident from the data presented- again- what in the paragraph starting line 210 is relevant for the main story?

Response: Yes, this section is relevant. Because the SacB signal sequence is still present, we needed to establish that the additional peptide would not interfere with scaffold formation, and that scaffolds are formed under the conditions that the protein building blocks are secreted by *B.*

subtilis. We confirmed that comparable structures are assembled than we had previously observed with purified EutM-SpyCatcher protein.

11. Figure 4b- nice if you like images of proteins and their active site- but- is it only relevant here if you discuss the behaviour of the different mutants in relation to their position/ accessibility? This is not done but could be.

Response: We agree that the functional effects of the different mutations could be discussed further based on their structural locations. But this would be beyond the scope of this manuscript though. However, by including their positions in the flagellin protein structure, we provide important information that will inspire future engineering of *B. subtilis* for protein surface display. We have added a statement to explain better our rational why we chose these positions: *“based on the assumption that mutations in these locations would least likely interfere with filament assembly and function”*.

12. Figure 4d- is this necessary in the main paper? You don't really make much reference to it in the text. Legends for this figure- far too long.

Response: Figure 4d serves as a quick guide to explain the fluorescence microscopy results in 4e and 4f. It shows the workflow of how we tested functional SpyTag display on flagella by binding to tdTomato-SpyCatcher. This figure also helps the reader to understand how the different cellular features and proteins were labeled. Importantly, it shows that not all flagellin subunits (black, genomic copy) contain a SpyTag insertion (yellow, plasmid borne copy). We therefore strongly recommend keeping Figure 4d.

13. Line 245 onwards. Use of the word 'some' 'best' 'shorter' all qualitative and not entirely convincing. If you have the data make it more robust. If you don't need really need anything other than a qualitative 'check' then say such and move on.

Response: Qualitative statements for phenotypic observation from microscopy data are commonly used to describe features. It is understood that they are descriptive. We found that the flagella have different morphologies that can be best described as short or curved as opposed to the helical WT flagella (see also additional images in Supporting Fig. 9).

14. Line 255. Explain why it is important that your construct can make 'isopeptide' bonds (think of your intended audience).

Response: We have added in brackets *“(for covalent attachment)”* to this sentence. We realized that the SpyTag-SpyCatcher system could benefit from additional explanation in the manuscript, especially for readers outside the genetic engineering, synthetic biology field. We therefore also introduced covalent isopeptide bond formation in the introduction and in the legend of Fig. 1.

15. Line 268. Ensure that Fig. S10a and related figures includes the His tag domain to be clear HOW the various components are connected. You allude to the His tag in the main text but the supplementary figure is missing information.

Response: We are not sure what the reviewer means here. The His-tag is shown in Fig. S10a. It is also shown in every schematic of the genetic constructs: Fig. 1, Fig. 2a and Fig. 5b. Figure 5b shows all of the genetic constructs used in this study. We now point out more explicitly in the main text that we compared silica precipitation by biomineralization peptide and unmodified His-EutM scaffolds (line 286, lines 290-91 in the revised manuscript).

16. Good to see that you use well controlled silicifying solutions though note- at the concentrations you are using condensation will start as soon as the precursor starts to hydrolyse- so, to be consistent with all of the excellent experimental descriptions- please include in your experimental the time (80 mins?) after the start of hydrolysis that you added your silicifying solutions to your cell media. This can be added to the SI.

Response: We have added sentence in the supporting information to clarify this: “*Under these conditions, hydrolysis is complete after 80 mins¹⁶ for use as silica source in silica binding and biomineralization experiments.*”

17. Fig. 5a, S10b- ‘rolled up tubes’ scale of images is not helpful to see this well- Energy dispersive x-ray analysis is needed to conform the presence of silica. From the scale of the images-it is not easy to see the increase in roughness you state in the text. You could describe the materials in terms of particle sizes- does silica addition affect this- HOW much? Does silica addition generate larger structures overall suggesting a coating or do the structures shrink? Even with the data you have you could say more that would be useful in terms of taking the materials forward. What properties of your composites will be useful?

Response: We believe the description of scaffold morphologies is of relevance as it confirms scaffold assembly and points out differences in assembly between the modified and unmodified scaffold building blocks, which would be expected upon fusion of charged peptide tags. The “rolled up tubes” in our print-out of Fig. 5a are visible for His-EutM without silica.

We tried but found it is not possible to make meaningful quantitative statements on the thickness of the silica coatings or size of the scaffolds based on comparing SEM images. The size of the scaffolds appears not to be significantly different before and after silica addition. If anything, they seem to have slightly shrunk upon coating with silica. A detailed characterization of the effect of silica precipitation on our protein scaffolds is outside the scope of this study, which is on creating living materials.

We used purified proteins to confirm that the fused peptide biomineralize silica and to select one peptide with the best biomineralization properties. The scaffold proteins are secreted and therefore need to function in solution for the formation of a living material. Most important for this ELM work was therefore to compare silica precipitation in solution by the modified scaffold proteins, for which we used the molybdate blue assay. Silica precipitation on scaffolds imaged by electron microscopy therefore served to confirm results obtained with the molybdate blue method. Conditions during cultivation are not the same as the controlled, and artificial conditions used for SEM or TEM though. But we have added supporting TEM imaging data (Fig. S10c) for silica precipitation by His-EutM and His-EutM-CotB coated TEM grids. Both scaffolds showed the deposition of a silica layer, which was more granular for EutM-CotB. Observations are included in the manuscript (see also response to reviewer 1).

EDX analysis of silica coating on His-EutM scaffolds. Scaffolds on TEM grids were incubated for 1 or 2 h with 100 mM silica. Red circles indicate location where Si/Cu signal was measured. The Cu signal derives from the underlying copper grit. The Si/Cu signal of the protein free area does not change significantly, while the Si/Cu signal on the scaffolds increases due to silica deposition. In addition, the presence of straight edges and the sharp angles of the silica coated region after 2 h suggest that silica deposition followed the underlying protein structure.

We agree that the scaffolds are interesting for the assembly of protein-based, functional biocomposite materials. In other work we are characterizing them for biomineralization applications and have done some preliminary EDX analysis (one example for His-EutM is shown to the right) with the goal of systematically quantifying silica coating on different scaffold types and engineered configurations. However, this is a separate study outside of the scope of this manuscript.

18. Line 305/ and or Figure S11 etc- add culture concentrations (however you want to refer to this- there is no information in the SI for these expts). Associated figure S12 etc- I did not find convincing. Is the section from Line 317 relevant to the main story?

Response: We have added the cultivation conditions and the final growth parameters (OD and pH) for Figure S11 as a new section in the Methods section of the Supporting Information. We believe that the experiments from Figure S11 and the associated section in the main text are relevant for establishing that secreted EutM-CotB enhances biomineralization under cultivation conditions (see also our comments above that confirmation of biomineralization in solution by protein scaffolds is important for ELM fabrication).

Supporting Fig. 12 shows cysteine labeled structures that are associated with cells only when EutM-SpyCatcher is expressed and secreted. It provides additional supporting evidence for scaffold secretion and co-localization with cells as shown in Fig. 7. It also demonstrates that the scaffold cysteine residues are accessible for dye labeling in future work. Other researcher may find that using this dye is a valuable method for visualization of secreted protein complexes.

19. Figure 6. S13 and associated Text. ‘Larger’ etc- measure and provide semi-quantitative data.

Response: We have measured the size of the aggregates as suggested (using ImageJ on raw images). Aggregates formed by cultures that display SpyTagged flagella for material cross linking are about twice the size compared to those that do not display a SpyTag. We have added this in the text as *“Larger silica material aggregates (about twice the size) were...”*

20. Rheological measurements- as a minimum the numbers require X plus/minus SD but you can test whether they are statistically significantly different and report the same in the abstract. Data can be used from Table S6.

Response: Means and standard deviations for rheology data from three biological replicates were included in Table S6. We performed additional statistical analysis (ANOVA, T-test) on these data as shown in Table S7 and statistical methods added to the Methods section. Except for the angular frequency data for the loss moduli (G'') of silica blocks generated from strains transformed with the control plasmid pCT-empty and pCT-EutMCotB-EutMSpyC, all other interactions are statistically significant ($p < 0.05$). The loss modulus (G'') was not a property affected by the secreting *B. subtilis* cultures. We added Table S7 and refer to it in the main text. As mentioned above, we had to shorten the abstract down to 150 words.it

21. The data to support the location of the secreted scaffolds 7a is convincing, that of 7b and images in the SI less so. I would have expected to see images at much higher magnification than presented here.

Response: As suggested by the reviewer, we have added a higher magnification image to show a close-up of the clear zones surrounding cell boundaries. This image is now provided in Supporting Fig.14a.

22. Line 380 onwards. A 24 hour sample is NOT dry and the silica component of the material will still be condensing. Again- you refer to ‘functional coatings or plasters’ ‘integrate desired functions’- what are these? Not clear how you do these experiments- do you break down the silica material- if not- do you remove it after a certain time? - how much ‘Si’ gets into solution etc before adding ‘more’ hydrolysed TEOS. A great pity that certain of the mutations do not survive to 2 weeks- though good to see you have a strategy to overcome this.

Response: We think the reviewer may have misunderstood the regeneration experiment. Silica blocks (material) were made as shown (and described in the text and methods) in Figure 6. Pieces of solid silica gel blocks were cut for thin section and TEM analysis (Figure 7). For regeneration, we took a small piece ($\sim 5 \text{ mm}^3$) from a block cured for 1 day (as described in the text) and inoculated fresh cultures. There would be an insignificant amount of silica added to the cultures from such a small piece. Cultures were then regrown and shown to retain their ability to secrete scaffolds and biomineralize silica in the same manner as the culture that originally generated the silica block from which the $\sim 5 \text{ mm}^3$ initial inoculum was derived.

As described above, we explained what is meant with “functional coatings and plasters” in the introduction. To clarify what we mean with “integrate desired functions” in line 403, we extended this sentence to “integrate desired functions through their living components”.

We agree that long-term stability of genetic information will be required for any true ELM. As we discuss in our manuscript, it is well known that plasmids in *Bacillus* are prone to mutation and are genetically unstable. This can however be overcome by chromosomal integration of the expression constructs that are currently residing on plasmids. But for engineering and optimization purposes, this is not practical until fine tuning of strain properties is complete. This is something that we will do in the future. But for us it was very encouraging to see that *B. subtilis*, even though it is dormant in the silica material, it retains the plasmids (albeit with mutation once stored for longer than 2 weeks).

23. Line 425. Be careful in your use of the word silica’ - you add a mix of silica oligomers/ polymers (depending on concentration used as a minimum) so please rethink whether this word is appropriate.

Response: We emphasized again that silica is derived from TEOS: “100 mM silica (derived from TEOS).” We are aware of the composition of silica and we therefore included early in the manuscript a reference (Belton et al. 2012) on silica mineralization.

24. In general, figures and legends need modifying- see examples above. Some further data analysis is needed to properly support some of the statements made.

Response: See our comment above. We have condensed legends when possible and included statistical analysis as requested.

Reviewers' Comments:

Reviewer #1:

Remarks to the Author:

Thank you - my comments have been addressed. I think that the introduction and rationale for this work will be clearer with the sentences added to the new version of the manuscript.

Reviewer #2:

Remarks to the Author:

The authors have addressed my only concern, and so I am happy that this great paper is now published. Well done to all involved

Reviewer #3:

Remarks to the Author:

I am happy with the changes made to the manuscript that improve clarity- this extends to the changes made to the figure legends and the material present in the SI. All improve readability of the paper and the potential to explore the data collected in this elegant study.

There is one small change to the abstract that is still needed- it is the second time 'silica material' is referred to - the change to include the fact that this material includes cells has been overlooked.

Please change.

Response to Reviewers Comments:

We thank the reviewers for their positive response to our revisions. We have made one minor change in the abstract in response to reviewer 3. We have indeed overlooked their request to mention that the silica material contains cells.

Reviewer #1 (Remarks to the Author):

Thank you - my comments have been addressed. I think that the introduction and rationale for this work will be clearer with the sentences added to the new version of the manuscript.

Reviewer #2 (Remarks to the Author):

The authors have addressed my only concern, and so I am happy that this great paper is now published. Well done to all involved

Reviewer #3 (Remarks to the Author):

I am happy with the changes made to the manuscript that improve clarity- this extends to the changes made to the figure legends and the material present in the SI. All improve readability of the paper and the potential to explore the data collected in this elegant study.

There is one small change to the abstract that is still needed- it is the second time 'silica material' is referred to - the change to include the fact that this material includes cells has been overlooked. Please change.

Response: We have added "*cell containing* silica material".